# Molecular-channel driven actuator with considerations for multiple configurations and color switching

Jiuke Mu[1], Gang Wang [2], Hongping Yan[3], Huayu Li[2], Xuemin Wang[4], Enlai Gao[5], Chengyi Hou[1], Anh Thi Cam Pham[4], Lianjun Wu[4], Qinghong Zhang[1,6], Yaogang Li[6], Zhiping Xu[5], Yang Guo[1], Elsa Reichmanis[2], Hongzhi Wang[1] & Meifang Zhu[1]

The ability to achieve simultaneous intrinsic deformation with fast response in commercially available materials that can safely contact skin continues to be an unresolved challenge for artificial actuating materials. Rather than using a microporous structure, here we show an ambient-driven actuator that takes advantage of inherent nanoscale molecular channels within a commercial perfluorosulfonic acid ionomer (PFSA) film, fabricated by simple solution processing to realize a rapid response, self-adaptive, and exceptionally stable actuation. Selective patterning of PFSA films on an inert soft substrate (polyethylene terephthalate film) facilitates the formation of a range of different geometries, including a 2D (two-dimensional) roll or 3D (three-dimensional) helical structure in response to vapor stimuli. Chemical modification of the surface allowed the development of a kirigami-inspired single-layer actuator for personal humidity and heat management through macroscale geometric design features, to afford a bilayer stimuli-responsive actuator with multicolor switching capability.

[1] State Key Laboratory for Modification of Chemical Fibres and Polymer Materials, College of Material Science and Engineering, Donghua University, 201620 Shanghai, China. [2] School of Chemical and Biomolecular Engineering, School of Chemistry and Biochemistry, School of Materials Science and Engineering, Georgia Institute of Technology, Atlanta, GA 30332, USA. [3] Stanford Synchrotron Radiation Light Source, SLAC National Accelerator Laboratory, 2575 Sand Hill Road, Menlo Park, CA 94025, USA. [4] Department of Mechanical Engineering, University of Texas at Dallas, Richardson, TX 75080, USA. [5] Applied Mechanics Laboratory, Department of Engineering Mechanics and Center for Nano and Micro Mechanics, Tsinghua University, 100084 Beijing, China. [6] Engineering Research Center of Advanced Glasses Manufacturing Technology MOE, Donghua University, 201620 Shanghai, China. Jiuke Mu and Gang Wang contributed equally to this work. Correspondence and requests for materials should be addressed to Q.Z. (email: zhangqh@dhu.edu.cn) or to E.R. (email: ereichmanis@chbe.gatech.edu) or to wanghz@dhu.edu.cn)

For many living things, self-adaptive movement in response to stimuli from exposure to sunlight, humidity, wetting, or other atmospheric conditions is often a necessity for survival[1,2]. Most of such motion derives from nonuniform internal structures that possess a range of dissimilarly oriented layers with different rigidity, expansion/contraction, or swelling properties. Learning from these biological systems, many artificial actuating materials that can generate a mechanical response from other forms of energy have been developed, and are well suited for applications such as microrobotics, sensors, actuators, and stealth materials[3–5]. The materials of construction for the fabrication of various vapor-driven actuators include polymer gels[6–8], elastomers[9], shape memory polymers[10], carbon nanomaterials[11–13], and electroactive polymers[14], which are not generally available on an industrial-scale and often lack environmental stability and biocapability. The absence of commercially available materials with product-level stability will constrain the implementation of actuators for applications in wearable devices for human–environment interface communications. Furthermore, currently reported actuation systems are driven by heat, pH, light, or electricity. Vapor-driven soft actuators are more desirable for those systems designed to allow the human body to better adapt to changing conditions; quite simply, the human body is always exchanging vapor with the environment.

For vapor-driven soft actuators, the two critical features that define performance are: (i) the incorporation of a vapor-absorbing functional group and (ii) a microstructure that facilitates the transport of vapor molecules. Polymers with an appropriate molecular structure and functional groups that are sensitive to vapor are readily accessible[5,15,16]. Furthermore, vapor-driven actuating materials have been developed through the design of gradient microporous or multilayer structures[17–20]. However, practical application of such systems has lagged significantly, owing to inherently complicated synthetic processes, slow kinematics associated with the requisite molecular organization, long response times caused by relatively slow vapor transport mechanisms, and poor machinability (see summary in Supplementary Table 1). Thus, developing a new generation of actuating materials with properties that commensurate with use under physiological conditions is an urgent need.

Nanoscale molecular channels exist widely in living cells and tissue and exhibit outstanding performance for storage and transport of small molecules[3]. The thigmonastic self-adaptive reaction of plants such as *Mimosa pudica* or the Venus flytrap (*Dionaea muscipula*) to external stimuli presents just one example. These plants lock energy in their leaves and form a trap using the pressure differential between two hydraulic layers[21]. After opening water channels and redistributing water molecules in response to the external stimulus, the latch is removed and the leaf relaxes to the new closed equilibrium state[22]. Molecular channels are found as key features in many science and engineering fields, including energy storage and environmental protection[23–25]. The molecular absorption/desorption inside a molecular channel has also been applied to the adaptive actuation of macroscopic films[26,27]. Thus, a self-adaptive, vapor-driven, and mechanically responsive system inspired by the molecular channels produced in nature can be envisioned.

The use of perfluorosulfonic acid ionomer (PFSA), otherwise known as Nafion™, as an ion-exchange membrane in ionic polymer–metal composite (IPMC) artificial muscle has been reported[28,29]. Actuation is induced by application of an electric field, whereby water molecules within the membrane move between two metal electrodes. IPMC performance, however, is limited due to electrolysis of water and/or water evaporation from cracks in the electrodes. Long-term cycling also leads to adhesion failure at the sandwiched interfaces.

Here we take advantage of the nanoscale molecular channels and vapor-absorbing functional groups present in Nafion™ to fabricate a series of vapor-driven actuators. The commercially available membrane material was studied as an intrinsically deformable and foundational material for a vapor-driven actuator. A series of vapor-driven single-layer and bilayer actuators that exhibited a rapid response (up to 0.25 s for a 75 μm single-layer PFSA film actuator under 18% ethanol vapor atmosphere) and high stability (> 8000 cycles without deterioration) were designed and fabricated by a simple and practical process,while avoiding interface problems and long-range movement of molecules in the nano channel. Multidimensional actuation, including two-dimensional bending, as well as three-dimensional helical actuation with twisting/untwisting and elongation/contraction deformation capability, was obtained through structural design of the interface. A kirigami-inspired single-layer actuator system was developed for personal humidity and heat management through the design of the macroscale geometry. Furthermore, chemical modification of the surface led to the development of a bilayer stimuli-responsive actuator with multicolor switching capability. This vapor-deformable microactuator is expected to have broad relevance in human-interface applications, e.g., personal humidity and heat management smart textiles, low observable technology, and artificial intelligence.

## Results

**Molecular structure and in-situ characterization of PFSA.** PFSA, an ionic polymer, is composed of a hydrophobic tetrafluoroethylene (TFE) backbone and hydrophilic sulfonic acid side chains[30,31]. It is generally accepted that the sulfonic acid groups cluster to form a hydrophilic microphase, dispersed throughout the continuous TFE phase[17,32]. The backbone TFE chains can be aligned to form straight helical crystalline regions, whereas branching TFE chains disrupt the alignment, thereby forming an amorphous region. The sulfonic acid groups ($-SO_3$) associated with both phases form randomly interconnected cylindrical nanochannels (schematic diagram in Fig. 1a)[33,34].

The channels, which are stabilized on the outside by relatively straight helical backbone segments, can absorb moisture or other polar vapor molecules from the environment when their concentration increases above a threshold value[35]. Upon exposure to a relatively high vapor concentration, the absorbed molecules diffuse into the hydrophilic domains at the membrane/vapor interface, causing the polymer to swell and change its crystalline structure. With a decrease in concentration, vapor molecules can escape, leading to shrinkage of the membrane and recovery of its crystalline structure. The absorption/desorption process (Supplementary Figure 1) was verified through in-situ analysis of the volume variations of the PFSA membrane. To further investigate the morphology evolution upon exposure to vapor, in-situ GIWAXS characterization was performed using ethanol as the polar medium (film thickness was controlled at ca. 400 nm). Specifically, ethanol was deposited onto the film, which was then monitored as it evaporated to leave a dry film (Fig. 1b, c). The video of the in-situ time-dependent 2D GIWAXS patterns and the reduced scattering intensities versus X-ray momentum transfer $q$ (defined as $q = 4\pi\sin\theta/\lambda$, in which $2\theta$ is the scattering angle and $\lambda$ is the wavelength of the incident X-rays) can be found in the supporting information (Supplementary Movie 1, Supplementary Figure 2 and Supplementary Figure 3). From these results, obvious scattering peaks owing to variations in the ordering of the film microstructure were observed. In comparison to the scattering peak observed for the dry film at $q \sim 1.2$ Å$^{-1}$, the deposition of ethanol led to the appearance of two major peaks at a $q$ of ~0.76 and ~1.6 Å$^{-1}$. The ~1.2 Å$^{-1}$ peak corresponds to the

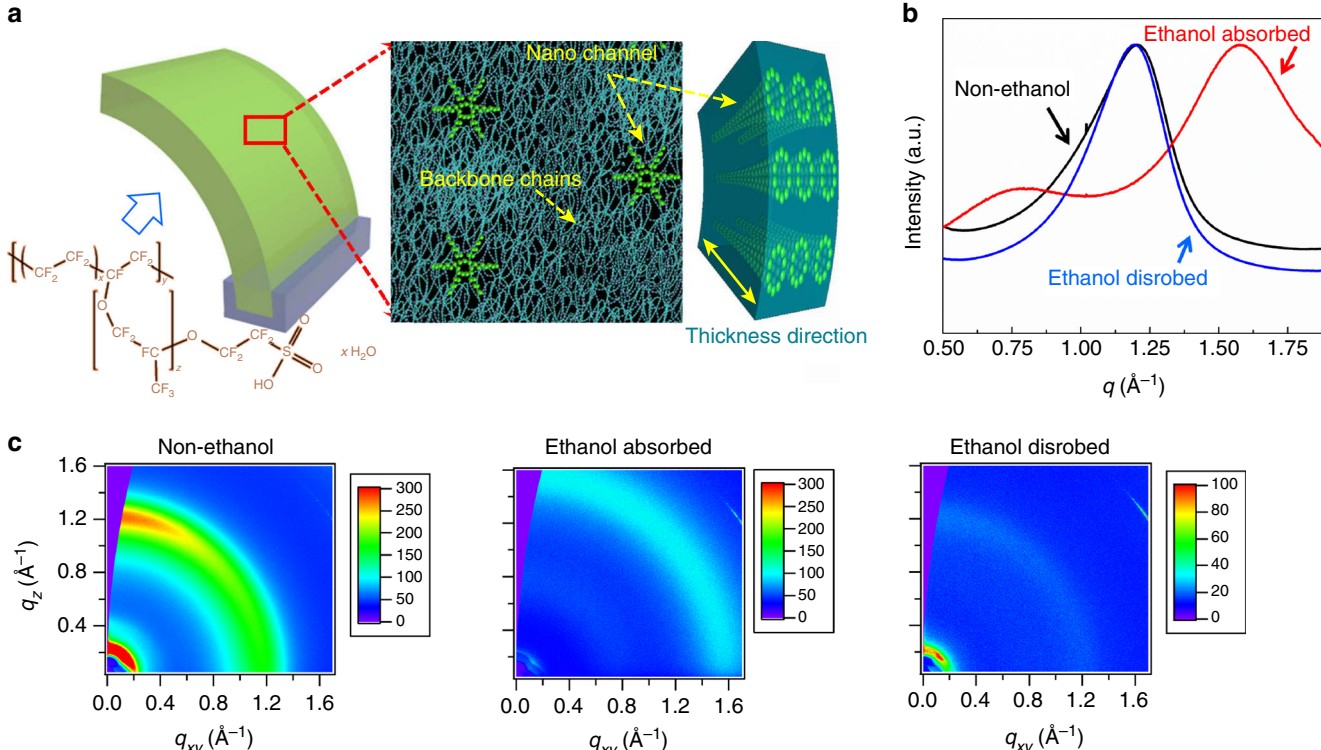

**Fig. 1** PFSA microstructure and in-situ GIWAXS characterization. **a** Chemical structure and microstructure with molecular channel distributions. **b**, **c** The GIWAXS patterns and corresponding line-cuts from the out-of-plane direction. The nominal thickness of the sample is 232 ± 17 nm

known ordering of the −CF₂− chains in the crystalline regions[30,36]. As ethanol evaporated from the membrane, the intensity of the ~1.6 Å⁻¹ scattering peak, associated with X-ray scattering from liquid ethanol, decreased as the film dried, with no apparent change in peak position during the process[37]. At the later stages of drying, the characteristic crystalline −CF₂− chain scattering peak at ~1.2 Å⁻¹ emerged, and its intensity increased, suggesting that the ethanol processed film recovered its original microstructure. The process appeared to be fully reversible.

The scattering peak at ~0.76 Å⁻¹, corresponding to a spacing of ~8.3 Å, is associated with ordering that results from incorporation of ethanol into the hydrophilic nanochannels. Over time, ethanol desorbed and the peak position shifted to ~0.85 Å⁻¹ (a decreased spacing of 7.4 Å). This change suggests that the absorption of ethanol introduced a volume expansion of ~40%, assuming that other regions of the film were not compressed (Supplementary Figure 3). Atomic force microscopy (AFM) imaging (Supplementary Figure 4), demonstrated that volume expansion did occur; the membrane was smooth initially, but became creased owing to swelling during vapor sorption. The circular edge of the stacked membrane layers was fixed. Supplementary Figure 5a presents engineering stress-strain curves of PFSA films (75 μm) exposed to different vapor conditions. The effects of the atmospheric conditions (ethanol vapor concentration and RH) on Young's modulus in both the machine and transverse directions are shown in Supplementary Figure 5b. The results indicate that higher ethanol vapor concentration and relative humidity lead to a decrease in Young's modulus.

**Single-layer membrane actuation mechanism.** Asymmetric volume expansion is known to induce bending of a film, and thus provides one means to induce actuation[18,38]. Integration of this mechanism with the hydrophilic/hydrophobic microphase-

separated structure of PFSA-based films suggests that it may serve as a promising candidate for vapor-driven actuators. The fast and reversible self-adaptive actuation of a PFSA single-layer membrane is presented in Fig. 2a, b. Exposure of one side of a PFSA 5 × 15 mm long membrane to ethanol vapor led to bending of the film; the bending cycle was 1.5 s, and a maximum curvature (using the minimum size of the radius to calculate the curvature, Supplementary Figure 6a and b) of 0.31 mm⁻¹ was observed at 0.25 s (for a 75 μm single-layer PFSA film actuator under the 18% ethanol vapor atmosphere, Supplementary Movie 2). To clearly demonstrate the level of actuation control, the change in bending curvature of single-layer PFSA films with different thickness upon exposure to 18% ethanol vapor conditions and relaxation dynamics after removing the ethanol vapor exposure is presented in Supplementary Figure 7a. The observed actuation performance and response time were comparable, or even superior, to the previous reports of typical vapor, solvent, light, as well as thermally driven actuators (Supplementary Figure 7b, and Supplementary Table 1). For a given PFSA thickness, the increased ethanol vapor concentration in air afforded an increase in PFSA membrane curvature (Fig. 2b inset); the absorption of ethanol increased as the concentration increased, and thus, the film experienced a larger volume expansion. In addition, the bending actuation performance of single-layer PFSA film also can be affected by gravity and stimulus direction. Specifically, as shown in Supplementary Figure 6c, d and 7c, the single-layer PFSA actuator film is always bent in the opposite direction to the higher vapor concentration side owing to the asymmetric volume expansion of the two sides of the actuator film, and the phenomenon is film thickness-dependent (Supplementary Figure 8).

The actuation process appeared robust—the cycle was performed several thousand times with no apparent fatigue, as presented in Fig. 2c. Attenuated total reflectance-infrared (ATR-IR) spectroscopy was used to monitor ethanol desorption

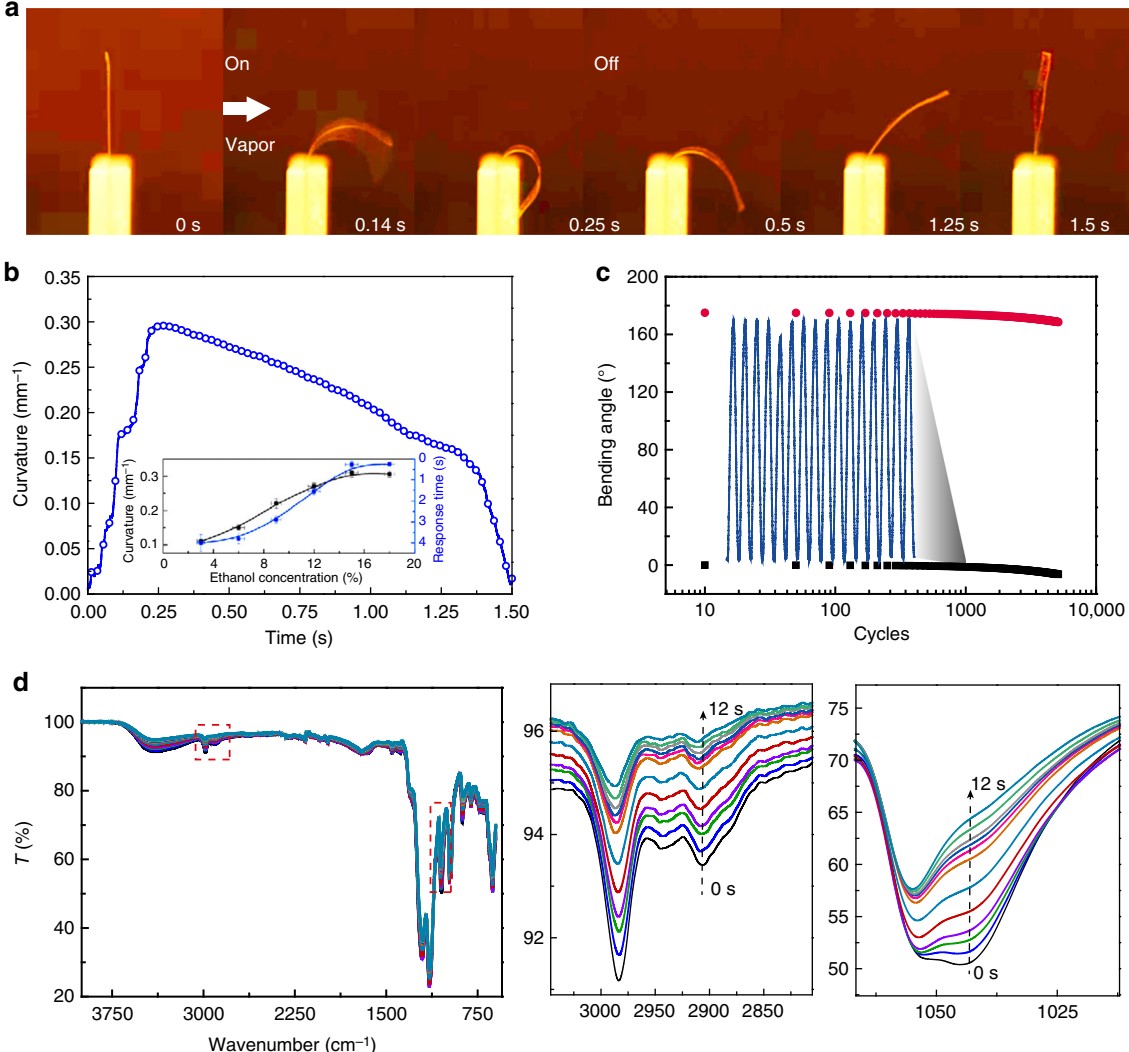

**Fig. 2** Actuation performance of the single-layer membrane. **a** Adaptive actuation movement of a PFSA membrane (5 mm × 15 mm × 75 μm) in response to ethanol vapor sorption and desorption (the vapor is from the left side of the PFSA membrane). **b** Plots of curvature against time (inset: a schematic of the curvature calculation). The insert image is the actuation curvatures and response time of a 75 μm PFSA film under different ethanol vapor concentrations. **c** Plots of the reversible deformation of the 75 μm film upon cyclic exposure to a 15% ethanol concentration vapor (8000 cycles without obvious fatigue). **d** Time-dependent ATR-infrared spectra of film saturated by exposing in ethanol vapor for 10 s (bottom to top: 0–12 s after exposing in air)

(Fig. 2d). Over time, the typical ethanol vibrations for −CH$_3$ and C−O stretching at 2910 cm$^{-1}$ and 1040 cm$^{-1}$, respectively, became weak, suggesting that ethanol could be absorbed in varying amounts and desorption occurred rapidly.

In comparison to vapor-driven actuators possessing a microporous structure, the molecular nano channel structure presents two significant advantages: first is rapid mass transfer induced by the nanofluidic channels[39,40]; second is a larger functionalized area that favors adsorption (Supplementary Note 2). These two features acted in concert, and thus a higher degree of atmospheric ethanol vapor was adsorbed, which provided for the rapid and exceptional range of actuation performance[18]. To elucidate the underlying vapor-driven mechanism, a Quartz crystal microbalance was used to measure the real-time weight change to determine the relationship between actuation and PFSA membrane quality change (Fig. 3a). The experimental apparatus is presented in Supplementary Figure 6a: a small hole in the wall of the vapor chamber (blue) allowed control of the internal ethanol vapor concentration ≈18%, while the other chamber (red) was maintained ethanol free through use of a N$_2$ flow. The absorption behavior of the membrane was readily determined from the

weight increase (Process i in Fig. 3a, b). In Process ii, the bending angle continued to increase due to inertia, whereas the weight remained constant owing to the osmotic pressure balance between the membrane surface and the external environment[41]. Upon removal of ethanol vapor, ethanol molecules were released from the channel and the membrane recovered its original shape (Process iii). On the other hand, if the sample continued to be exposed to ethanol vapor, the bending angle of the PFSA film decreased partly, but the film did not return to the flat state (Fig. 3c). Comparing these two situations, the degree to which the PFSA film decreased (Fig. 3c) is close to the observed increase in Process ii (Fig. 3b). Thus, it can be inferred that the PFSA film behavior in Process ii is due to inertia.

The degree of nanochannels expansion was determined from the amount of absorbed vapor molecules. As shown in Fig. 3d, the channels were initially in a closed state. They gradually expanded as more vapor molecules were absorbed. The expansion ratio of the side of the membrane exposed to vapor was larger. The interfacial stress induced by the unbalanced expansion of the two surfaces introduced bending actuation.

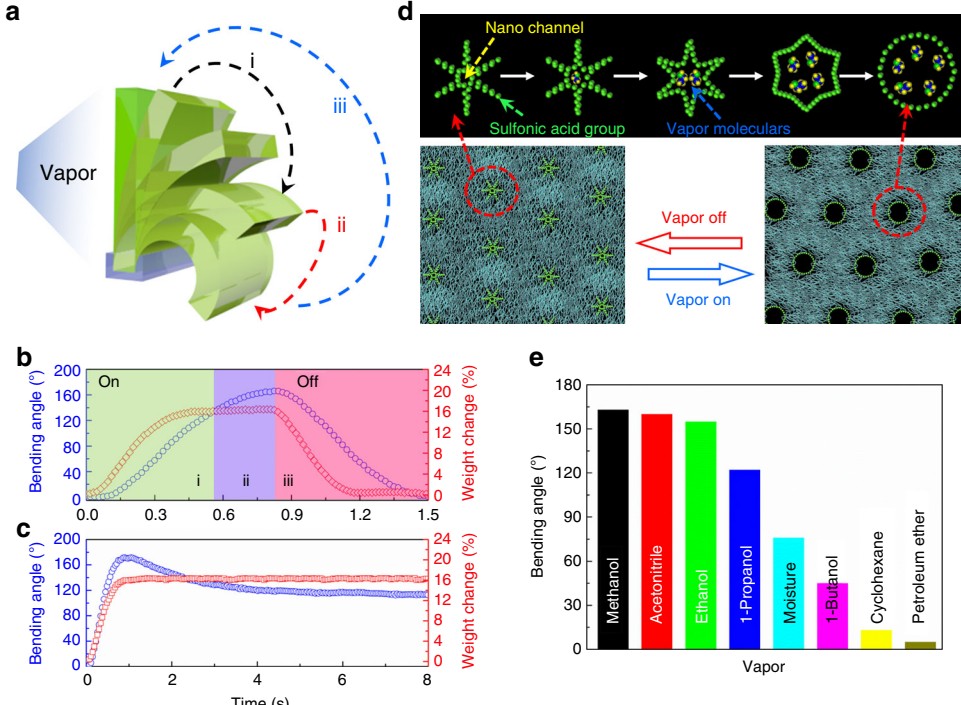

**Fig. 3** Actuation mechanism of the nano channel membrane. **a** Schematic representation of the vapor-responsive bending principle of the actuator (when vapor approaches one side of the single-layer membrane, the molecular channel adsorbs the vapor and the surface undergoes swelling, resulting in the film bending upon exposure to the vapor source). **b** Single-layer membrane bending and reversing actuation synchronizes with weight change when the vapor was pumped in and out. **c** Single-layer membrane bending behavior with weight change when the vapor was retained in the chamber. **d** Schematic representation of the molecular channel expansion process. **e** Maximum bending angles of the single-layer PFSA actuator triggered by various solvent vapors. (2 kPa, 20 °C)

Multi-responsive bending actuation with both moisture and alternative solvent vapors was demonstrated (Fig. 3e). We found that high polarity and high saturated vapor pressure solvent vapors (i.e., methanol, ethanol, acetonitrile) (Supplementary Table 2), that is, vapors that interact strongly with the nano channel surface, trigger the strongest actuation; solvent vapors (i.e., cyclohexane, petroleum ether) that have low polarity drive a much weaker bending response. In particular, the polymer film, of which the two sides are exposed to different concentrations of ethanol, can be treated as a problem of diffusion through a thin plate. When a steady state is reached, a concentration gradient of ethanol induces a swelling gradient along the thickness direction, which apparently leads to the bending behavior. This behavior can be characterized by the curvature, and the relationship between the swelling gradient and curvature is given theoretically as Supplementary Eq. (1):

$$1/\rho = \alpha\left(c^0 - c^L\right)/L, \tag{1}$$

in which $\rho$ is the curvature of the single-layer actuator, $\alpha$ is the swelling coefficient, $L$ is the thickness of the membrane, $c$ is the relative vapor molecular concentration across the membrane (Supplementary Note 1, Note 2, Supplementary Figure 9a and b), and validated by both experiment and finite element analysis. The curvature under these gradient strains was calculated both from the simulations and Eq. (1), as presented in Supplementary Figure 9; the results are consistent with each other. Furthermore, we used both the experiments and simulations to interrogate the contribution of the effect of the clamped position and the structure of actuators (length-width ratio) to bending direction (Supplementary Figure 10a, b and c).

**Self-adaptive and multifunctional bilayer actuators**. One approach to achieving anisotropic motion in artificial actuators uses a bilayer system, in which the active layer is composed of a vapor-responsive material in conjunction with an inert material. Here, an effective strategy was developed to produce multiple forms of humidity-driven actuation (from two-dimensional spreading to three-dimensional helical buckling). As the active layer, a PFSA solution mixed with rhodamine B was sprayed onto the surface of a flexible inert layer (polyethylene glycol terephthalate (PET)). A schematic representation of the fabrication procedure is shown in Supplementary Figure 11a. Upon solvent evaporation, and as evidenced by cross-sectional scanning electron microscopy (SEM) images (Supplementary Figure 11b), the PFSA layer was in intimate contact with the underlying PET substrate, and the initially flat bilayer film became tubular in shape, owing to the internal stress between the two dissimilar materials (Supplementary Figure 11c). When moisture was pumped onto the surface of the curled bilayer film, swelling of the thinner PFSA layer resulted in the complete spreading of the relatively thicker bilayer film (Supplementary Figure 11d and Supplementary Movie 3). Stopping the moisture flow caused the bilayer film to recoil back to its original state. The active-to-inner layer thickness ratio can be used to control the curvature of the tube, both before and after spreading (Fig. 4a). The trend associated with actuation curvature increased and then decreased as the thickness ratio increased. Simulations were performed to explain the spreading direction and how the thickness ratio affects actuation. When two layers are bonded together, but expand unequally, there is a natural tendency for the composite to spread. The active layer has a higher expansion coefficient toward moisture than the inert layer, and thus the active layer will stretch more. As a result, the composite will spread.

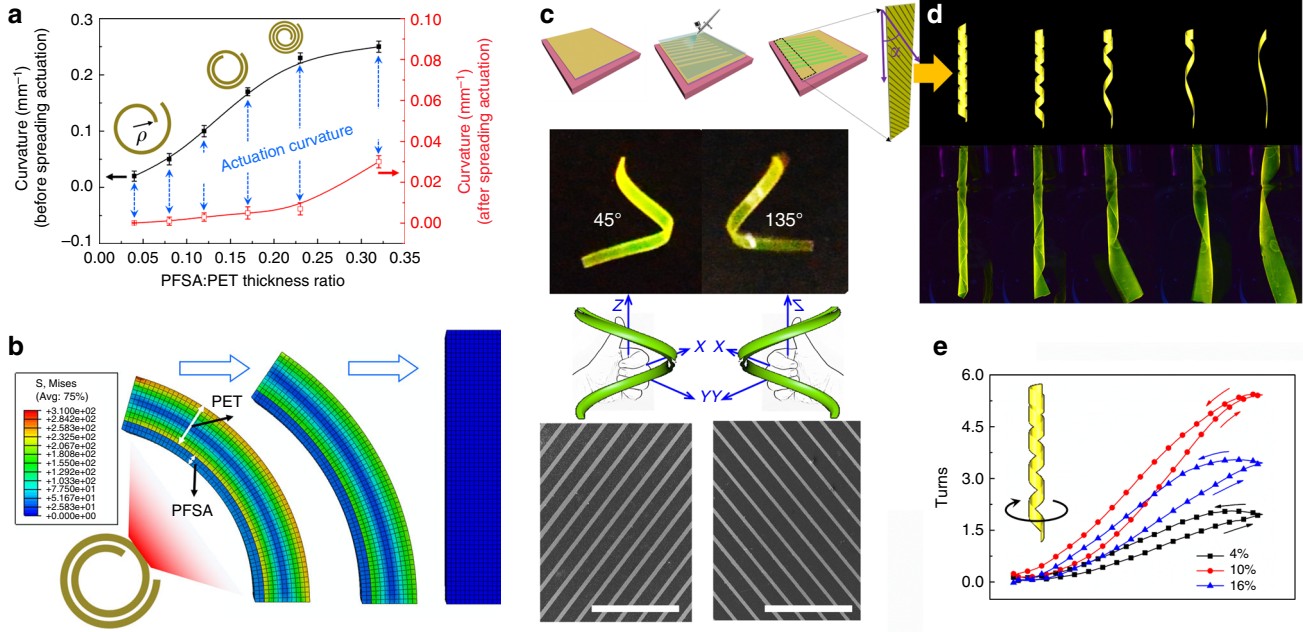

**Fig. 4** PFSA-based helical actuation. **a** Plots showing the bending curvature for the 20 × 60 mm bilayer actuator depending on the ratio of the active-to-inner layer thickness. **b** FEA results of stress distribution associated with the bending actuation of bilayer actuator. **c** The scheme illustrating the preparation of the three-dimensional helical buckling with an active PFSA layer line. Photos and schematic illustration of the right-handed and left-handed deformation of the composite strips with an $\alpha$ of 45° and 135°. SEM images of aligned PFSA lines. (Scale bar is 400 μm). **d**, **e** Schematic illustration, photographs, and time-dependent untwisting actuation of the helical strip before and after triggering by moisture

Theoretical simulations were carried out using Abaqus[42] to calculate the curvature change when applying the equivalent moment (Fig. 4b)[43]. The simulation curvature change is consistent with the theoretical result

$$\rho = \frac{1}{R} = \frac{a(t_2 - t_1)}{h_p}\left(\frac{6mn(1+m)}{1 + 4mn + 6m^2n + 4m^3n + m^4n^2}\right) \quad (2)$$

Here, $\rho$ is the curvature of the composite, $h_a$ and $h_p$ is the thickness of the active and passive layer, respectively, $E_1$ and $E_2$ is Young's modulus of the active and inert layers, respectively, $m = h_a/h_p$, $n = E_a/E_p$, and $\alpha$ is the expansion coefficient of moisture of active layer. Humidity increases from $t_1$ to $t_2$. The simulation results confirm the experimental observations (Fig. 4a and Supplementary Figure 12). Experimentally, incorporation of rhodamine B facilitated direct observation of the actuation process (Supplementary Figure 11c and d).

Furthermore, by manipulating the orientation and geometric shape of the active PFSA layer, multiple modes of three-dimensional helical buckling actuation were developed. Upon solvent volatilization, contractile stress perpendicular to the orientation of an aligned PFSA line led to a reversible transition of the patterned sheet to a helix. Notably, the morphology of the helix was suitably controlled by varying the angle between the PFSA line orientation and the longitudinal direction of the strip ($\alpha$); not only right-handed, but also left-handed helical deformations were generated at 45° and 135° (Fig. 4c). Based on this tunable helical structure, increased humidity led to a rapid untwisting transformation accompanied by elongation. The structure remained in this tensional state until the humidity returned to its original state (Fig. 4d, e, and Supplementary Movie 3). The demonstration of facile PFSA-based actuation suggests applications in heat management and humidity sensing. These are briefly demonstrated below.

## Kirigami-inspired self-adaptive actuator system for personal humidity and heat management.

The moisture-driven bending actuation that derived from PFSA membranes motivated the design of a macroscale geometric, kirigami-inspired design of a personal humidity and heat management system. As presented in Fig. 5a, direct laser machining was used to fabricate heat management systems with a range of different patterns. As shown in Fig. 5b and Supplementary Figure 13, (double) semilunar/square-pattern matrixes were simultaneously cut and patterned on the surface of the actuators. After patterning, the PFSA membrane was attached to an opening of a controlled humidity chamber. With increased humidity, the membrane patterns curled simultaneously in a direction external to the chamber, with a concomitant increase in the dimensions of the open channels separating the internal and external chamber environment (Fig. 5b, Supplementary Movie 4 and Supplementary Movie 5). These channels facilitated the control of humidity between the two environments. In the absence of the patterned PFSA membrane, the humidity within the enclosed chamber could be increased continuously to reach saturation, ca. 100%; however, when the membrane was affixed to an opening in the chamber, the humidity within the chamber reached an equilibrium state of ca. 40% and remained constant (Fig. 5c).

The ability of the PFSA-patterned membrane to exhibit actuation upon changes in humidity suggests that the approach might be applicable to personal humidity and heat management systems. For instance, it is well known that exercise and heat generally lead to increased levels of perspiration in humans. To evaluate the efficacy of the patterned PFSA actuator, a PFSA matrix with semilunar patterns able to curl in an outward direction was integrated into a commercial sports shirt to develop clothing for personal humidity and heat management, as shown in Fig. 5d, e. The performance of the actuator-based sports shirt was compared with that of a standard terylene-based sports shirt. Water vapor transmission rate (WVTR) tests were conducted on an unpatented PFSA film-based shirt, PFSA film shirt having

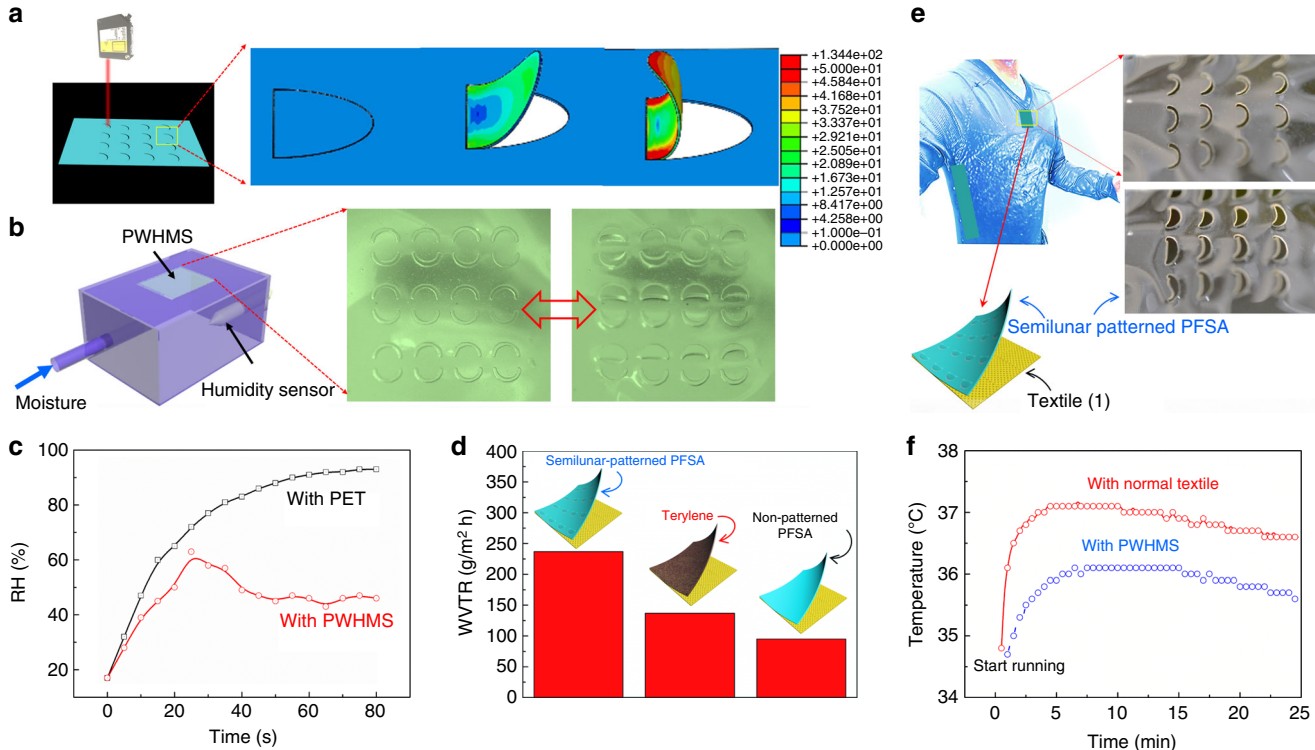

**Fig. 5** Personal humidity and heat management system. **a** Fabrication of the actuator array using a laser cutting and patterning system. **b** Schematic diagram of the homemade humidity chamber used for quantitative measurement of the humidity control ability of personal humidity and heat management system. The photo of the actuator array before and after self-adaptive actuation. (Scale bar is 1 cm). **c** Plots of time-dependent humidity change within the chamber affixed with normal PET film and personal humidity and heat management system. **d** Water vapor transmission rate test of three type of PFSA films with semilunar patterns shirt, standard polyester fabric sports shirt, and PFSA film (without any pattern)-based shirt. **e**, **f** A PFSA matrix with semilunar patterns able to curl in an outward direction was integrated into commercial sports shirt to develop clothing for personal humidity and heat management. Plots of the time-dependent skin temperature. (Scale bar is 1.5 cm)

semilunar patterns and a standard polyester fabric sports shirt at RH = 90%. The data in Fig. 5d demonstrate that the standard polyester fabric shirt exhibited a higher WVTR than the unpatented PFSA analog, 145 versus 97 g/m$^2$/h, respectively. The actuator-based sports shirt exhibited a significantly higher WVTR, up to 237 g/m$^2$ per hour. Furthermore, a test that involved a man running 3 km was also conducted. When wearing the PFSA actuator-based shirt, the runner's skin temperature was as much as 1.3 °C lower than that when wearing typical commercially available sportswear (Fig. 5e, f). Thus, the PFSA layer appears to enable venting between the skin and the external environment, thereby facilitating control of body moisture and skin temperature.

**Interactive mechanochromic actuation (color switching) through PFSA surface modification.** Integration of chromogenic photonic crystals, such as SiO$_2$ microspheres, with intrinsically deformable soft materials, can lead to stimuli-responsive mechanochromic materials that exhibit multicolor switching over the visible region. The PFSA films described above afford a platform to demonstrate such mechanochromic actuation, whereby the system possesses both interactive color-switching and humidity-sensing properties.

As shown in Fig. 6a, a flexible multilayer composite film was fabricated by partially silver plating both sides of a PFSA film, followed by coating the plated area with well-ordered SiO$_2$ nanoparticles encapsulated by PDMS (Supplementary Figure 14). The resultant film underwent bending actuation associated with the uncovered, nascent PFSA segment (Fig. 6b), exhibiting a range of deformations as measured in degrees, upon changes in

relative humidity. With a constant viewing angle, the composite, multilayer film segment (covered by silver plating and SiO$_2$ nanoparticles/PDMS) changed color when the degree of bending either increased or decreased with respect to neutral.

Reflection spectroscopy was used to characterize the mechanochromic behavior by recording the spectral evolution versus time as relative humidity increased (Fig. 6b, c). Bragg's equation was applied to elucidate the displayed color:

$$\lambda = 2dn_{\text{eff}} = (8/3)^{1/2}D(0.74n_{\text{p}}^2 + 0.26n_{\text{m}}^2 - \sin^2\varphi)^{1/2} \qquad (3)$$

Here, $\lambda$ is the wavelength of the reflected color, $D$ is the particle diameter, $n_{\text{p}}$ is the refractive index of the particles, $n_{\text{m}}$ is the refractive index of the matrix, $\varphi$ is the angle between the incident beam and the (1 1 1) direction of the face-centered cubic (fcc) stacking, $d$ is the (1 1 1) plane spacing, and $n_{\text{eff}}$ is the effective refractive index. Changes to the inter-scattering distance ($D$, $n_{\text{p}}$, or $d$) and the refractive index of the materials ($n_{\text{m}}$ or $n_{\text{eff}}$), which are known to tune the photonic properties of colloidal systems, were not evaluated[15,44]. According to Bragg's law, an fcc colloidal crystal of SiO$_2$ particles ($n_{\text{p}} = 1.45$, $n_{\text{m}} = 1$) with a diameter of 375 nm should exhibit a reflection peak at $\lambda = 612(1.81−\sin2\varphi)^{1/2}$, for different viewing angles (incident light), which coincides with the various reflection colors. To further investigate the color hue of the actuator, the color coordinates were calculated and displayed (black symbols) on the coherent infrared energy chromaticity diagram presented in Fig. 6d. The results confirmed that the color of the film could be readily adjusted by changing the humidity.

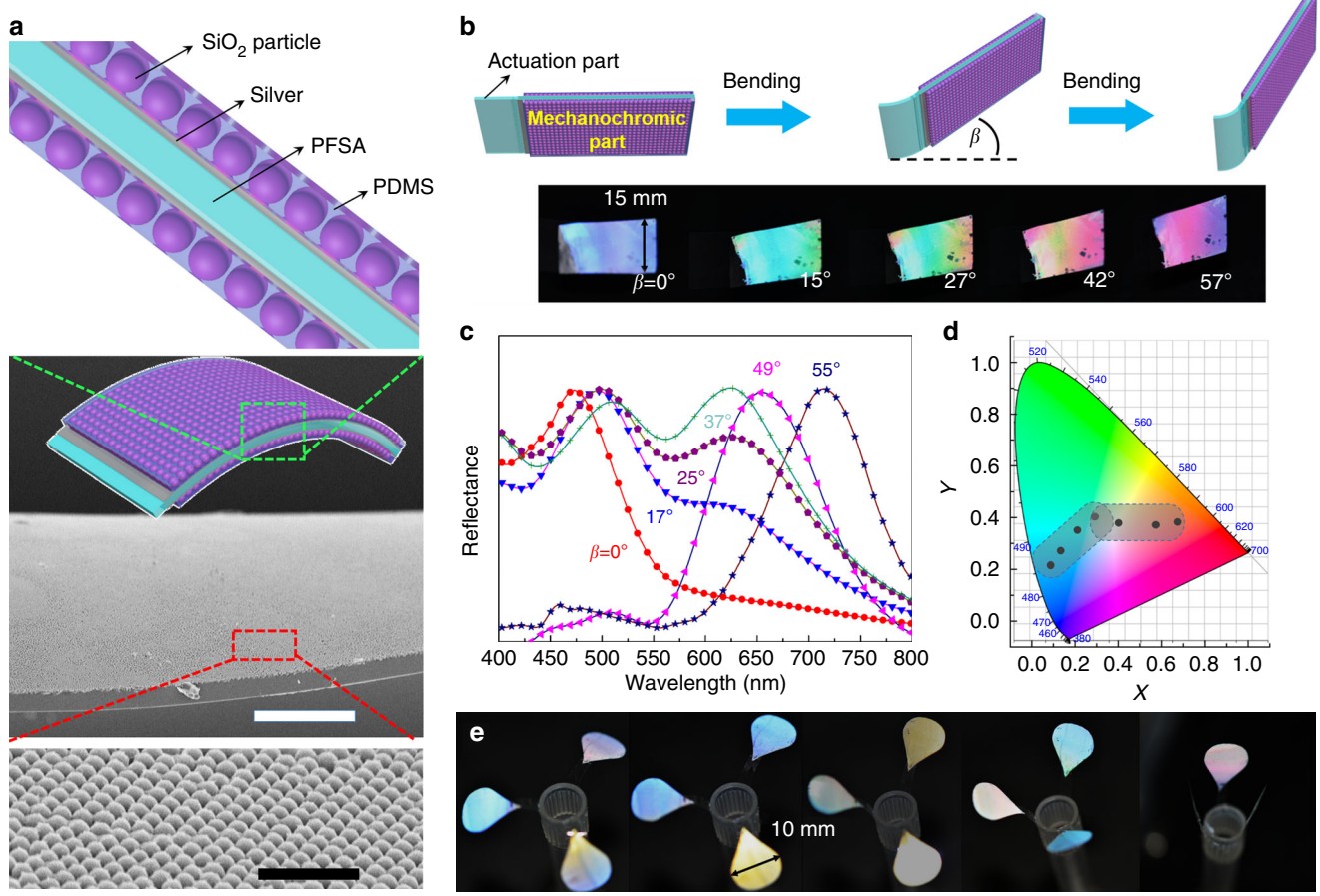

**Fig. 6** Stimuli-responsive color change actuator. **a** Schematic illustration of the cross-sectional structure of the stimuli-responsive color change actuator, and SEM images of a colloidal crystal layer. **b** Schematic illustration and photograph of the stimuli-responsive color change actuator. The composite film exhibited a range of bending deformity and changed color with an increase in relative humidity. **c** Measured angle-resolved specular reflection of the color change actuator. In the specular reflection measurements, the incident light is normal to the sample surface. **d** The color coordinates were calculated and displayed (black symbols) on a CIE-1931 color space chromaticity diagram. **e** A smart flower fabricated by stimuli-responsive color change actuator, which can change its shape and color depending on the humidity

Furthermore, the response time was only 1.2 s, a value that is faster than most previous reports[16,45].

Owing to the relatively wide range of achievable colors, rapid response time, and large actuation amplitude, the PFSA-based system was used to demonstrate smart color-changing flowers, as shown in Fig. 6e, in the which color was determined by external humidity. As a somewhat whimsical application, the interactive mechanochromic smart actuator might also serve as an invisibility cloak imagined in Harry Potter and the Prisoner of Azkaban (Supplementary Figure 15)[46]. As an example, at a relative humidity of ca. 17%, a piece of the material might be very distinct relative to its surroundings because of its color; increasing the humidity to ca. 29% leads to actuation and resultant deformation with an apparent color change. A further increase in humidity would afford added changes such that the PFSA composite simply fades into the background creating the invisibility cloak. One might then imagine a range of applications—from whimsical childrens' toys to humidity sensors to technologies that can manipulate observability.

## Discussion

Inspired by the molecular channels that widely exist in living cells and tissue, a conceptually simple, multifunctional, vapor-driven actuator exhibiting a fast response time was prepared using a commercially available PFSA film. Absorption (or desorption) of polar vapors (water, alcohols, and acetone) into (from) PFSA films leads to film deformation as determined by time-dependent in-situ GIWAXS. The in-situ GIWAXS characterization suggested that the microstructures of the PFSA films are reversible during the deformation process. With an interface design, deformation in multiple dimensions was demonstrated, from two-dimensional bending actuation to three-dimensional helical actuation with twisting/untwisting and elongation/contraction deformation capability, exhibiting extremely rapid response (0.25 s for one actuation) and ultrahigh stability (>8,000 times without deterioration). These results point to a range of possible applications. In particular, a kirigami-inspired single-layer actuator was developed as a personal humidity and heat management system through a macroscale geometric design, and a double-layer stimuli-responsive actuator with multicolor switching capability was achieved through chemical modification of the surface. The intrinsically deformable PFSA soft actuators represent an excellent platform for the future development of smart materials for human–environment interface applications.

## Methods

**Preparation of the single-layer actuator**. For the preparation of the PFSA membrane, PFSA solution (5%, Aldrich) was cast on a glass plate and dried (80 °C, 2 h) in a vacuum oven. After drying, a free-standing membrane was easily peeled off from the glass plate. The AFM was used to demonstrate the surface structure have been provided in Supplementary Figure 5c and d.

**Preparation of interactive bilayer actuators**. The PFSA solution (5%) mixed with rhodamine B (mass fraction 1%) was sprayed on the surface of flexible polyethylene glycol terephthalate and dried under ambient conditions. Along with the solvent evaporation, the PFSA layer was in close contact with the underlying PET substrate, as clearly shown by the SEM images of an actuator cross section (Supplementary Figure 11b).

**Preparation of an interactive mechanochromic actuator**. Utilizing vacuum evaporation plating technology detailed in Supplementary Figure 14, a 50 nm-thick silver film was applied to promote the brightness of the structure color[41]. Subsequently, $SiO_2$ microspheres with a typical diameter of 375 nm were deposited onto the $O_2$-plasma-treated silver layer through the sol-dip coating. The appearance of the shinning colors indicated that the $SiO_2$ microspheres rapidly assembled into the well-ordered fcc structure with the evaporation of the solvent (shown in the cross-sectional SEM images in Fig. 6a). Finally, after encapsulation of the $SiO_2$ layer with PDMS through a dip-coating method, a multilayer composite film was fabricated.

**Characterization and measurements**. The in-situ and ex-situ GIWAXS experiments were performed at Stanford Synchrotron Radiation Lightsource beamline 11–3 with X-rays of energy 12.7 KeV using a Rayonix MX225 CCD area detector. An incident angle of 0.14° was used to probe the bulk morphology of the films. For the in-situ experiment, the sample was first aligned, and a drop of ethanol was deposited on top, followed by a collection of the GIWAXS image data at a rate of 10 s per frame. The data were analyzed using personalized Igor Pro code based on NIKA package and personalized code by Dr. Stefan Oosterhout. The morphologies of the as-prepared samples were determined by SEM (JSM-6700F, JEOL, Tokyo, Japan), and photographs were taken using a single-lens reflex camera (D7000, Nikon). AFM images were recorded using an AFM (Nanoscope IV SPM, Digital Instruments). A laboratory balance (AL204, Mettler Toledo) was used to collect mass data. Temperatures and IR thermal images were recorded using an IR thermometer (FLIR A40M, Thermo-Vision). The bending angle was measured using a laser displacement sensor (KEYENCE IL-030). ATR-IR spectra were recorded on a Nicolet NEXUS-670 spectrometer.

**Data availability**. The data sets generated and/or analyzed in this study are available from the corresponding author on reasonable request.

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

## Acknowledgements

We gratefully acknowledge the financial supports by NSF of China (No. 51672043), MOE of China (No.111-2-04, IRT_16R13), STC of Shanghai (16JC1400700, 15ZR1401200, and 16XD1400100), SMEC (2017-01-07-00-03-E00055), and Eastern Scholar. ER appreciates

support from the Georgia Institute of Technology, the Brook Byers Institute for Sustainable Systems, and the Center for Science and Technology of Advanced Materials and Interfaces. Use of the Stanford Synchrotron Radiation Light Source, SLAC National Accelerator Laboratory, was supported by the U.S. Department of Energy, Office of Science, and Office of Basic Energy Sciences under contract No. DE-AC02-76SF00515.

## Author contributions

J. M. and G.W. contributed equally to this work. J.M, H.W., E.R., Q.Z., and G.W. conceived the idea, designed the research, and carried out the experiments. E.G., X.W., H.L., Z.X., L.W., H.Y., and A.T.C.P. performed the numerical simulation. M.Z., H.Y., and C.H. performed the mechanical fatigue experiments. J.M., Q.Z., M.Z., and Y.G. analyzed and interpreted the results. The manuscript was written through the contributions of all authors. All authors have given approval to the final version of the manuscript.

## Additional information

**Competing interests:** The authors declare no competing financial interests.

