## [Peer Review File · Nature Communications]

Reviewers' comments:

Reviewer #1 (Remarks to the Author):

In this manuscript, the authors studied the actuation performance of perfluorosulfonic acid ionomer (PFSA) film under vapor stimulus, and further designing various configurations for interesting applications in multidimensional space and biomimetic system. However, there are still some major questions or concerns to be addressed as below. The mechanism parts need to be further studied and the manuscript is not suitable for publication in its current form.

1. The author emphasized that the PFSA is commercially available materials in the abstract and introduction, what is the advantage?
2. In Line 97 (Page 4), it is debatable for the description "A series of vapor driven monolayer and bilayer actuators that exhibited a rapid response (0.25 s for one curl)", because that the response time should be variable depending on the thickness of film, structure of actuator and the vapor conditions.
3. The mechanical properties of the film under vapor condition should be provided to the reader due to it is important for the following practical applications.
4. As author explained, the vapor adsorption will make the volumes changes of the materials, whether the length changes under ethanol vapor triggering when the thickness changes?
5. In the time-dependent 2D GIWAXS patterns, the desorption of ethanol from the membrane needed about tens of second, why the response time of the actuator is about 0.25s?
6. In the Figure S4, the explanation of colors of AFM results should be detailed.
7. In Line 160 (Page 7), the reference should be more comprehensive. For example, many carbon based actuators also exhibit fast response time, but not be included here.
8. The curvature should be calculated carefully due to it is not homogeneous in different length of film as shown in Figure 2a.
9. The actuation of film should be demonstrated when the film is fixed in different angles to horizontal plane because of the effect of gravity force.
10. In Line 171, the author explained the advantage of molecular nanochannel structure compared with macroporous structure, is there any other data supporting this?
11. In Figure 3a, the film become bend with the absorption of vapor. In Process ii, the bending angle continued to increase when the vapor remained constant. However, As I understand it, the bending angle should decrease because of the homogeneous absorption of vapor after that the film was exposed to the vapor for a lone time.
12. In the preparation, the film is in vacuum oven. Whether the film wrinkled when being exposed to air because of the existence of water molecules.
13. The precise content of final film should be given.
14. The performance of actuator should rely on the vapor condition (e.g. speed, density).
15. The volume change should be uniform along with all direction, why does the film bend to a specific direction in all actuators showed in the manuscript?

.....

Reviewer #2 (Remarks to the Author):

In this manuscript, the authors demonstrated the vapor-driven actuation of perfluorosulfonic acid ionomer (PFSA) and its utilization in three impressive applications: 2D/3D bilayer actuator, wearable actuator for humidity/heat management, as well as interactive mechanochromic actuation with the integration of chromogenic photonic crystals. These applications are uniquely designed, well demonstrated and undoubtedly may enable several new applications. This manuscript should be considered for publication after the following modifications:

1. Differentiation: PFSA film, such as DuPont NAFION (same chemical structure as presented in Figure 1A) is known for its nanochannel structure for ion transportation, and actuator applications, such as ionic polymer-metal composite (IPMC) has drawn many interests. Although the focus of IPMC is switching electrical signals to shape formation, the bending mechanism is similar and has been extensively studied. The authors should discuss those applications in the introduction session with a clear explanation of the novelty of the work presented in this manuscript.

2. Monolayer PFSA actuation property experimental: For sensor/actuator application, reproducibility and selectivity are important figure of merits. I would recommend to present Fig. S5b, S6 in the main content, possibly in combination with Fig. 2. In addition, from Fig. S5a, the PFSA membrane only experienced vapor exposure from one side. If this is true, the discussion of actuation performance simulation (supplement note 3, Fig. 3c, Fig. S7) is confusing as the vapor concentration on the opposite side of the film (cL) is undefined and will continuously increase with time (i.e., not a steady state). The authors should elaborate more, possibly includes a diffusivity and time-dependent error function, in order to find better agreement between the simulation results (Fig. 3c, Fig. S7, up to 5% of swelling strain gradient) and the experimental results (Figure 2b and Fig. 3a as transient condition and Fig. S5 as pseudo-steady-state condition, up to 160 degree of bending angle which is much larger in magnitude compared to 5% of swelling strain gradient).

Reviewer #3 (Remarks to the Author):

The authors present interesting results from research on vapor-driven, polymeric thin film actuators. The novelty here is the demonstration that PFSA (Nafion) thin films, which intrinsically contain nanoscale molecular channels, can respond quickly to changes in the humidity or ethanol vapor concentration with large conformational changes. Nafion thin films have been used in many sensing and actuation applications, but typically are actuated by application of a potential across the film. What's novel and surprising here is the rather fast and repeatable actuation by gradients in RH or other polar (e.g., ethanol) vapors. The authors argue that the ready availability of large sized PFSA thin films makes them interesting for integration into wearable textiles. They show that shirts with vapor/humidity actuated venting features can be fabricated and used for humidity and heat management of the wearer.

To this reviewer at least, the demonstration of self-adaptive bilayer actuators and interactive mechanochromic actuation were scientifically more exciting. As such, the authors should consider to change the title of the paper to focus more on these applications (which are not targeted to human-environment interface applications), and as an additional feature, also discuss the "heat and moisture regulating" shirt.

The paper is clearly written, and the data shown support the conclusions drawn. A few issues should be addressed however.

- 1) As mentioned, consider shifting the focus of the paper toward the surprising conformational responses of PFSA due to changes in humidity and other polar vapors.
- 2) The authors may wish to call out the PFSA also by its commercial name Nafion.
- 3) Page 3, Line 58: references should be added after each materials class.
- 4) Page 6: The thickness of the films studied by GIWAXS should be given.
- 5) Page 8: Provide an explanation why methanol yielded the largest bending actuation. What do we know about the specific interactions of methanol with PFSA?
- 6) Page 12: Was the standard terylene-based sports shirt also the same material used for the "actuated" sports shirt? The comparison between the shirts needs to be made more clear. For example, one could measure vapor diffusion (in absence of mechanical actuators), ideally it should be equal for both types of shirts to arrive at a reasonable comparison. This reviewer found this section the scientifically weakest, while the idea and implementation certainly are intriguing.

Response to comments

Reviewer 1

In this manuscript, the authors studied the actuation performance of perfluorosulfonic acid ionomer (PFSA) film under vapor stimulus, and further designing various configurations for interesting applications in multidimensional space and biomimetic system. However, there are still some major questions or concerns to be addressed as below. The mechanism parts need to be further studied and the manuscript is not suitable for publication in its current form.

We thank the Reviewer for finding our work interesting for “applications in multidimensional space and biomimetic system”. In this work, a new actuator that takes advantage of the intrinsic nanoscale molecular channels within a commercial perfluorosulfonic acid ionomer (PFSA) film was fabricated by simple solution processing to realize a self-adaptive and ambient-driven actuation. These findings should help many research groups in their studies related to soft actuators, such as “interactive mechanochromic actuation”, and “personal humidity and heat management”. This molecular-channel driven actuator is expected to have broad applications in human-environment interface applications.

According to the reviewer’s suggestions, additional statements/clarifications have been added to the Introduction and Conclusions to clarify the novelty and significance of this work (see details in the following section). More discussion about the in-situ GIWAXS characterization, deeper finite element analysis, and further experimental and theoretical investigations about the parameters that affect the actuation properties have been provided to further elucidate the mechanism. All the questions and concerns have been addressed, point-by-point. We hope that the current version can be published in Nature Communications.

1. *The author emphasized that the PFSA is commercially available materials in the abstract and introduction, what is the advantage?*

Response: Flexible polymer-based vapor driven actuators have been the subject of many research and development efforts for several years because of their many promising applications, including solvent sensors, humidity monitoring, soft robotics, and so forth. The realization of such vapor-driven actuators requires ease of fabrication, high actuation performance, and commercial availability, all of which are important for both fundamental research and practical applications. While systems based upon polymer gels, liquid crystal polymer and electroactive polymers have been widely reported, fabrication of environmentally stable and biocompatible vapor-driven actuators is difficult to achieve on an industrial-scale. The lack of commercial availability with commensurate product-level stability constrains their practical implementation. This work, which takes advantage of commercially available PFSA, widely used in fuel cells, sensors and super acid catalysis for the production of fine chemicals, paves a new strategy for wearable applications. The

discussion regarding the advantages associated with using a commercial material has been provided in the Introduction in Page 3 in the Revised Manuscript.

“The materials of construction for the fabrication of various vapor-driven actuators include polymer gels⁶⁻⁸, elastomers⁹, shape memory polymers¹⁰, carbon nanomaterials¹¹⁻¹³ and electroactive polymers¹⁴, which are not generally available on an industrial-scale and often lack environmental stability and bio-capability. The absence of commercially available materials with product-level stability will constrain the implementation of actuators for applications in wearable devices for human-environmental interface communications.” (Page 3, Line 58~64)

2. In Line 97 (Page 4), it is debatable for the description “A series of vapor driven monolayer and bilayer actuators that exhibited a rapid response (0.25 s for one curl)”, because that the response time should be variable depending on the thickness of film, structure of actuator and the vapor conditions.

Response: We appreciate your valuable suggestions. In the revised manuscript, “monolayer” was changed to “single-layer” to accurately describe the actuation film. In the original manuscript, the response time we mentioned was the quickest response time for the 75 μm single-layer PFSA actuator. In order to make the description more accurate, the statement concerning response time has been clarified in the Revised Manuscript. Furthermore, the effects of film thickness, actuator structure and the vapor conditions on the bending curvature and speed were quantitatively measured to clearly demonstrate the level of actuation control.

Revisions made to the manuscript and Supplementary Information include the following:

“We demonstrate that PFSA films curl/flatten upon absorption/desorption of polar vapors. A rapid response (up to 0.25 s for a 75 μm single-layer PFSA film actuator under 18% ethanol vapor atmosphere) and exceptional stability (>8000 cycles without deterioration) was achieved.” (Page 2, Line 34~37)

“To clearly demonstrate the level of actuation control, the change in bending curvature of single-layer PFSA films with different thickness upon exposure to 18% ethanol vapor conditions and relaxation dynamics after removing the ethanol vapor exposure is presented in Fig. S7a. The observed actuation performance and response time were comparable or even superior to previous reports of typical vapor, solvent, light, as well as thermally driven actuators (Fig. S7b, and Table S1).” (Page 7, Line 168~173)

Fig. S7 (a) Plots showing time dependence of the bending curvature change of single-layer PFSA films with different thickness upon exposure to 18% ethanol vapor conditions and relaxation dynamics after removing the ethanol vapor exposure. (b) Comparison of bending actuation properties and the response time in our study and previous bending actuators¹²⁻²³.

Table S1 Summary of different materials and structure for the design of actuators. Note: In each cited paper, we choose the most rapid speed for comparison with our results

Ref.	Response time (s)	Material (Structure)	Type of actuation
1	0.4	PILF ₂ N (Microporous)	Bending
2	0.38	Cross-linked LC polymers	Bending
3	2	Agarose	Bending
4	3	PVDF/PVA (Microporous)	Bending
5	6	PIL-PAA@tissue paper membranes (Microporous)	Bending
6	50	PEI/PAA film (Microporous)	Bending
7	15	Polyamide-6 substrate and a liquid-crystalline polymer	Bending

		coating	
8	1s	Carbon nanotube	Tensile
9	2.7	Carbon nanotube	Torsional
10	98s	Carbon nanotube	Torsional
11	50	Graphene oxide	Bending

3. *The mechanical properties of the film under vapor condition should be provided to the reader due to it is important for the following practical applications.*

Response: We do agree with the reviewer that the mechanical properties of the films under vapor conditions are important for subsequent practical applications. According to the reviewer's suggestions, new experiments have been conducted to investigate the mechanical properties under vapor conditions:

A new figure, Figure S5, and related discussion has been provided in the Revised Manuscript and Supporting Information.

“Fig. S5a presents engineering stress-strain curves of PFSA films (75 μm) exposed to different vapor conditions. The effects of the atmospheric conditions (ethanol vapor concentration and RH) on Young's modulus in both the machine and transverse directions are shown in Fig. S5b. The results indicate that higher ethanol vapor concentration and relative humidity lead to a decrease in Young's modulus.” (Page 7, Line 152~156)

Fig. S5 (a) Engineering stress-strain curves of PFSA films under conditions of no-vapor, 18% ethanol vapor and RH (relative humidity) = 90%. (b) Young's modulus as a function of ethanol vapor concentration and RH. (c, d) AFM images of freshly prepared PFSA films exposed to ambient air.

4. As author explained, the vapor adsorption will make the volumes changes of the materials, whether the length changes under ethanol vapor triggering when the thickness changes?

Response: Thank you for this question. In order to calculate the volume change precisely, the vapor-responsiveness of the PFSA was examined by observing the changes in their dimensions with an ethanol vapor trigger. As shown in the experimental apparatus (Figure S1a), a small piece of PFSA film was subjected to incremental increases in ethanol vapor concentration and changes in film dimensions were recorded by video and still micrographs. As shown in Fig. S1a, the dimensions of the PFSA film increased rapidly when ethanol vapor was introduced on its surface. In addition to checking the size change of PFSA film by top view, the side view was also recorded. These results demonstrated that ethanol vapor adsorption led to increases in both thickness and length/width. In Figure S1b, we plot the percentage of volume change of the PFSA film as a function of ethanol vapor concentration, film volume increased ~16% for exposure to an atmosphere of 18% ethanol vapor.

The revision made in the revised manuscript and revised Supplementary Information:

“The absorption/desorption process (Fig. S1) was verified through in-situ analysis of the volume variations of the PFSA membrane.” (Page6, Line 127~128)

Fig. S1 (a) Optical microscope images of a piece of PFSA film ($450 \times 750 \mu\text{m}$) which shows the changes in length and width of PFSA films in the dry state, and upon exposure to 18% ethanol vapor. (b) The equilibrium volume change of the PFSA films when exposed to different ethanol vapor conditions.

5. In the time-dependent 2D GIWAXS patterns, the desorption of ethanol from the membrane needed about tens of second, why the response time of the actuator is about 0.25s?

Response:

The response time for the thin film actuator is indeed 0.25 second according to the actuation experiment to test their response time. As noted above, the response time is affected by the vapor conditions, the film thickness, and even gravity. Here, for the time-dependent 2D GIWAXS patterns, in order to further understand the changes in domain ordering during ethanol actuation, we designed the in-situ X-ray scattering experiment so that the ordering spacing change during the “de-actuation” process can be clearly monitored in real time. The sample was placed in a stage for x-ray scattering, which was in a small chamber ($\sim 10 \times 10 \times 10 \text{ cm}^3$) with a Kapton window for X-rays passing through. Although the x-ray detector used in this experiment has a maximum image acquisition rate of 1 second/frame, we observed that the exposure time for each frame needed to be at least 10 seconds to have a sufficient signal-to-noise ratio. In order to allow ethanol evaporation at an appropriate rate for this exposure time, we controlled the opening of the sample chamber and the flow of the helium gas into

the chamber (which helped reducing air scattering of the X-rays). Hence, the evaporation and desorption of the ethanol are slowed down, which allowed us to monitor the de-actuation process. This process, however, did not reflect the response time of the actuator. An excess amount of ethanol was deposited manually on top of the pre-aligned sample film (on a silicon substrate), so the initial evaporation of the ethanol liquid allowed following experiment procedures (locking the X-ray hutch and starting image acquisition). Unfortunately, during this procedure, actuation happens immediately after ethanol deposition, so that fast process could not be monitored to ascertain the time-dependent microstructure evolution.

Text to clarify this point has been added to Page 3 in the Revised Supplementary Information:

“In order to further understand the change in domain ordering during ethanol actuation, we designed a time-dependent 2D GIWAXS, in-situ X-ray scattering experiment so as to monitor the ordering spacing change during “de-actuation” in real time. The sample was placed on an x-ray scattering stage, which was in a small chamber ($\sim 10 \times 10 \times 10 \text{ cm}^3$) fitted with a Kapton window to facilitate X-ray exposure. The x-ray detector used in this experiment has a maximum image acquisition rate of 1 second/frame, however, the required sample exposure time for each frame for sufficient signal to noise needed to be at least 10 seconds. In order to effectively modulate the ethanol evaporation rate to be commensurate with this exposure time, we controlled the opening of the sample chamber and the flow of the helium gas into the chamber (which also helped reduce scattering of the X-rays from residual air). The evaporation and desorption of the ethanol were thereby slowed down, which allowed us to monitor the de-actuation process. This process, however, did not reflect the true, as measured actuator response time.” (Page 3 of Supplementary Information, Line 42~55)

6. In the Figure S4, the explanation of colors of AFM results should be detailed

Response: The color scale associated with the AFM results in Figure S4 was used to visualize the surface height change of the PFSA membranes in going from the ‘off’ and ‘on’ states. The color scale bar used here is a parameter provided in the AFM analysis software package “Nanoscope Analysis”.

The explanation of colors of the AFM results has been clarified with more detail in the caption of Figure S4 in the Revised Supporting Information.

Fig.

S4 *In-situ* AFM height images of the PFSA membrane in the off state (without ethanol vapor evaporation) and on state (with ethanol vapor absorption). (The scanning area is $5\ \mu\text{m} \times 5\ \mu\text{m}$.)

7. In Line 160 (Page 7), the reference should be more comprehensive. For example, many carbon-based actuators also exhibit fast response time, but not be included here.

Response: Many thanks for your suggestions. More comprehensive references, including many carbon-based actuators, have been added in the Revised Manuscript and Supporting Information.

“The observed PFSA membrane actuation performance and response time were comparable with or even superior to those of typical vapor, solvent, light, as well as thermal driven actuators (Fig. S7b, Table S1).” (Page 8, Line 171~173)

8. The curvature should be calculated carefully due to it is not homogeneous in different length of film as shown in Figure 2a.

Response: Thank you for your suggestion. In this work, the position of the actuator which has a maximum curvature for every actuation moment was used to calculate and define the curvature of the bending actuation moment. In order to clarify the method, we provide a schematic diagram of experimental apparatus used for quantitative measurement of the curvature during film deformation in Fig. S6. As shown in Fig. S6b, to locate the sample, the right edge of the sample film was fixed by the holder, and the initial film was vertical to the ground. Three dashed circles (red, yellow and blue) represent the profile of the PFSA film. Here, “ r ” is the radius of every bent arch. Obviously, we chose the red circle which had the

minimum radius to calculate the curvature. The curvature was calculated by the equation below. $Curvature = 1/r$

The manuscript and Supplementary Information have been revised accordingly:

“Exposure of one side of a PFSA 5×15-mm long membrane to ethanol vapor led to bending of the film: the bending cycle was 1.5 s, and a maximum curvature (using the minimum measured radius to calculate the curvature, Fig. S6b) of 0.31 mm^{-1} was observed at 0.25 s (for a $75 \text{ }\mu\text{m}$ single-layer PFSA film actuator under 18% ethanol vapor, Movie S2).” (Page 7, Line 164~168)

Fig. S6 Effect of ethanol vapor concentration on the curvature of single-layer PFSA membrane actuator (a) schematic illustration of the experimental measurement apparatus. (b) The scheme used for definition and calculation of curvature. (The three dashed circles (red, yellow and blue) represent the profile of the PFSA film. Here, “ r ” is the radius of every bent arch. The curvature was calculated by the equation below. $Curvature = 1/r$)

9. The actuation of film should be demonstrated when the film is fixed in different angles to horizontal plane because of the effect of gravity force.

Response: Thank you for the very constructive suggestion! The actuation performance is tightly related to the film thickness. According to the valuable suggestions, the measurements obtained for the bending actuation of single-layer PFSA films has been provided. Films with three different film thicknesses, $45 \text{ }\mu\text{m}$, $70 \text{ }\mu\text{m}$, and $95 \text{ }\mu\text{m}$, were fixed at 8 different angles (0° , 45° , 90° , 135° , 180° , 225° , 270° , 315°) to the horizontal plane. We found that gravity certainly affects bending actuation performance, and the phenomenon is film thickness-dependent. For instance, the calculated value of bending curvature changed by more than 25% when $45 \text{ }\mu\text{m}$ thick films were fixed at different angles. However, as the film thickness increased, the impact of gravity on the bending curvature decreased typically to less than 5% percent when the thickness of the PFSA film actuator was $95 \text{ }\mu\text{m}$. With this information, the thickness of the film actuator can be tuned to the requirements of a given application.

A new figure to summarize the effects of gravity has been provided as Figure S8 in the supporting information and the related description has been added on Page 10 of the Revised Supporting Information.

Fig. S8 Max curvature of single-layer PFSA films with different thicknesses versus angles to which the film was fixed relative to the horizontal plane. The right image presents a scheme providing the definition of θ .

“Here, Films with three different film thicknesses, 45 μm , 70 μm , and 95 μm , were fixed at 8 different angles (0° , 45° , 90° , 135° , 180° , 225° , 270° , 315°) to the horizontal plane. We found that gravity certainly affects bending actuation performance, and the phenomenon is film thickness-dependent. For instance, the calculated value of bending curvature changed by more than 25% when 45 μm thick films were fixed at different angles. However, as the film thickness increased, the impact of gravity on the bending curvature decreased typically less than 5% percent when the thickness of the PFSA film actuator was 95 μm .” (Page 10 of Supporting Information, Line 135~142)

10. In Line 171, the author explained the advantage of molecular nano channel structure compared with microporous structure, is there any other data supporting this?

Response:

Reply: There are two significant advantages of the molecular nano channel structure compared with microporous analogs: (1) rapid mass transfer induced by the nanofluidic channels (*Adv. Mater.* 2017, DOI: DOI: 10.1002/adma.201702419; *Chem. Soc. Rev.* 2017, 46, 5400-5424); and (2) the larger functionalized area that favors adsorption of gas molecules. From prior reports, rapid mass and heat transfer can be facilitated by the introduction of nano-scale channels. Compared with the mass and heat transfer induced by micro-scale channels, nano channel structures provide for much faster vapor molecule

transfer which in turn provides for faster response time. For simplicity, it can be probed by a model in Note 2.

To clarify the importance of the molecular nano channels, the description has been revised on Page 9 of the Revised Manuscript and Page 13 of Supporting Information:

“In comparison to vapor-driven actuators possessing a **microporous** structure, **the molecular nano channel structure presents two significant advantages: (1) rapid mass transfer induced by the nanofluidic channels^{39,40}; (2) a larger functionalized area that favors adsorption (Note 2). These two features acted in concert, and thus a higher degree of atmospheric ethanol vapor was adsorbed, which provided for the rapid and exceptional range of actuation performance¹⁸.**” (Page 8, Line 184~188)

“Note 2: Comparison of the structure of films with different size of channels,

First, we assume the film has the same volume fraction of channels

$$\varphi = N\pi d^2 h / 4V \quad (3)$$

where N is the number of channel with diameter d and length h , and V is volume of the film.

The total interfacial functionalized area and channel length can be derived as

$$A = N\pi dh = 4V\varphi/d \quad (4)$$

and

$$L = Nh = 4V\varphi/\pi d^2 \quad (5)$$

Hence, the smaller d is, the larger A and L are. Furthermore, the larger functionalized area is in favor of adsorption of a gas molecule, and the higher surface-to-volume ratio is good for fast diffusion. Hence, these two factors play important roles in fast and large actuation associated with the molecular nano channel structure.” (Page 13 of Supporting Information, Line 183~194)

11. *In Figure 3a, the film become bend with the absorption of vapor. In Process ii, the bending angle continued to increase when the vapor remained constant. However, as I understand it, the bending angle should decrease because of the homogeneous absorption of vapor after that the film was exposed to the vapor for a lone time.*

Response: To explain this point, it is first necessary to recall the experimental apparatus for this measurement. As shown in Fig. S6a, we employed a vapor chamber (blue) with a small

hole to adjust the internal ethanol vapor concentration $\approx 18\%$, while the other chamber (red) was under a continuous flow of N_2 that enabled depletion of ethanol from the surrounding regions. For Process ii, although the weight of the PFSA actuator remained constant owing to the osmotic pressure balance between the membrane surface and the external environment, the bending behavior of the film could not stop immediately due to inertial forces.

In the case where vapor was not pumped out from the blue chamber, the bending angle of the PFSA film decreased, but it did not return to the original flat state (Fig. 3b bottom). Comparing these two situations, the degree to which the PFSA film decreased (Fig. 3b bottom) is very close to the degree to which the PFSA film increased bending in Process ii (Fig. 3b top). These results strongly suggest that the behavior of the PFSA film in Process ii is due to inertial forces. The bending angle remained stable (Fig. 3b bottom) because the vapor adsorption on the vapor source side and the desorption of the opposite side of the PFSA film finally reached a dynamic balance.

The following revision was made to the manuscript and Supplementary Information:

“To elucidate the underlying vapor driven mechanism, a Quartz crystal microbalance was used to measure the real-time weight change to determine the relationship between actuation and PFSA membrane quality change (Fig. 3a). **The experimental apparatus is presented in Fig. S6a: a small hole in the wall of the vapor chamber (blue) allowed control of the internal ethanol vapor concentration $\approx 18\%$, while the other chamber (red) was maintained ethanol free through use of a N_2 flow.** The absorption behavior of the membrane was readily determined from the weight increase (Process i in Fig. 3a and b). In Process ii, the bending angle continued to increase **due to inertia**, whereas the weight remained constant owing to the osmotic pressure balance between the membrane surface and the external environment⁴¹. Upon removal of ethanol vapor, ethanol molecules were released from the channel and the membrane recovered its original shape (Process iii). **On the other hand, if the sample continued to be exposed to ethanol vapor, the bending angle of the PFSA film decreased partly, but the film did not return to the flat state (Fig. 3b bottom). Comparing these two situations, the degree to which the PFSA film decreased (Fig. 3b bottom) is close to the observed increase in Process ii (Fig. 3b top). Thus, it can be inferred that the PFSA film behavior in Process ii is due to inertia.**” (Page 8, Line 188~203)

Figure 3 Actuation mechanism of the nano channel membrane. (a) Schematic representation of the vapor-responsive bending principle of the actuator (When vapor approaches one side of the single-layer membrane, the molecular channel adsorbs the vapor and the surface undergoes swelling, resulting in the film bending upon exposure to the vapor source). (b) Single-layer membrane bending and reversing actuation (top) synchronizes with weight change when the vapor was pumped in and out. Single-layer membrane bending behavior with weight change when the vapor was retained in the chamber (bottom). (c) FEA results of strain distribution associated with bending actuation of the single-layer membrane and schematic representation of the molecular channel response to different solvents (2 kPa, 20 °C). (d) Bending angles of the single-layer membrane exposed to various driving solvents.

12. *In the preparation, the film is in vacuum oven. Whether the film wrinkled when being exposed to air because of the existence of water molecules.*

Response: Thank you for this question. As we mention in the material preparation part, after the films were fabricated by a solution casting method, the PFSA film was peeled off from the glass. This peeled off process was completed in an oven which was maintained at low humidity (Relative Humidity= 5% ± 2). Further, fundamental to the vapor molecule absorption mechanism, the humidity conditions for both sides of the PFSA film were identical. Thus, both sides of the PFSA film would absorb the same amount of the moisture, and therefore both will swell to the same degree. As a result, there were no obvious wrinkles on the surface of the PFSA film upon exposure to air. Furthermore, AFM imaging was used to check the surface structure of a PFSA film. No obvious wrinkled structure could be identified.

The AFM images to demonstrate the surface structure have been provided in Fig. S5c and d in the Revised Supporting information. A related description to clarify this point has been added in the Revised Manuscript.

“The AFM images to demonstrate the surface structure of the PFSA films have been provided in Fig. S5c and d, which demonstrated that no obvious wrinkled structure could be identified.” (Page 7, Line 156~158)

13. The precise content of final film should be given.

Response: Information relating to the content of the final films has been provided in Figure 6a. To clarify this point, the related description about the precise content of final films has been provided on Page 14 in the Revised Manuscript.

“As shown in Fig. 6a, a flexible multilayer composite film was fabricated by partially silver plating both sides of a PFSA film followed by coating the plated area with well-ordered SiO₂ nanoparticles encapsulated by PDMS (Fig. S14). The resultant film underwent bending actuation associated with the uncovered, nascent PFSA segment (Fig. 6b), exhibiting a range of deformations as measured in degrees, upon changes in relative humidity. With a constant viewing angle, the composite, multilayer film segment (covered by silver plating and SiO₂ nanoparticles/PDMS) changed color when the degree of bending either increased or decreased with respect to neutral.” (Page 13, Line 311~317)

Figure 6 Stimuli-responsive color change actuator. (a) Schematic illustration of the cross-sectional structure of the stimuli-responsive color change actuator, and SEM images of a colloidal crystal layer (b) Schematic illustration and photograph of the stimuli-responsive color change actuator. The composite film exhibited a range of bending deformity and changed color with an increase in relative humidity. (c) Measured angle-resolved specular reflection of the color change actuator. In the specular reflection measurements, the incident light is normal to the sample surface. (d) The color coordinates were calculated and displayed (black symbols) on a CIE-1931 color space chromaticity diagram. (e) A smart flower fabricated by stimuli-responsive color change actuator, which can change its shape and color depending on the humidity.

14. *The performance of actuator should rely on the vapor condition (e.g. speed, density).*

Response: In the original manuscript and supporting information we provided the performance of two types of actuators (including a single-layer and a bilayer actuator) that rely on the ethanol vapor concentration (Fig. 2b insert and Fig. 4e). Here we provide more detailed data of actuation performance triggered by different vapor conditions.

In the original manuscript and supporting information, the vapor condition (e.g. speed, density) -dependent results were provided, please see original Fig. S5, along with related discussion: “As expected, an increased concentration of ethanol vapor in air afforded an increase in PFSA membrane curvature (Fig. S5); the absorption of ethanol increased when the concentration of the vapor increased, and thus, the volume expansion of the film was larger.” To better address the reviewer’s comments, in addition to the bending curvature, the bending response time triggered by different ethanol vapor concentrations is now also provided in Fig. 2b (insert). Furthermore, the bending actuation performance triggered by different vapors, which have different polarities was also included in Fig. 3d.

The revision made to the manuscript and Supplementary Information follows:

“For a given PFSA thickness, increased ethanol vapor concentration in air afforded an increase in PFSA membrane curvature (Fig. 2b insert); the absorption of ethanol increased as the concentration increased, and thus, the film experienced a larger volume expansion. In addition, the bending actuation performance of single-layer PFSA film also can be affected by gravity, and the phenomenon is film thickness-dependent as shown in Fig. S8.” (Page 7, Line 173~178)

“Multi-responsive bending actuation with both moisture and alternative solvent vapors was demonstrated (Fig. 3d). We found that high polarity and high saturated vapor pressure solvent vapors (i.e., methanol, ethanol, acetonitrile) (Table. S2), that is, vapors that interact strongly with the nano channel surface, trigger the strongest actuation; solvent vapors (i.e., cyclohexane, petroleum ether) that have low polarity drive a much weaker bending response.” (Page 9, Line 209~213)

Table S2. Polarity and saturated vapor pressure of solvents.

Solvent	Polarity	Saturated vapor pressure (kPa)
Methanol	6.6	16.8
Ethanol	4.3	8.5
1-Butanol	3.7	0.82
1-propanol	4	5.8
water	4.2	3.16
Acetonitrile	6.2	12.15
Cyclohexane	0.1	13.3
Petroleum ether	0.01	53.3

15. *The volume change should be uniform along with all direction, why does the film bend to a specific direction in all actuators showed in the manuscript?*

Response: Thank you for this question. As we understand the mechanism, the bending directions were attributed to the direction of the stimulus, the structure of the actuators and their clamped position. Firstly, both the single-layer and bilayer actuator PFSA actuator film always bent in the opposite direction to the stimulus owing to the asymmetric volume expansion of the two sides of the actuator film. Specifically, for the single-layer PFSA actuator film, we used both the experiments and simulations to interrogate the contribution of the effect of the clamped position and the structure of actuators (length-width ratio) to bending direction. As shown in Fig. S10a and b, the unclamped PFSA film actuator showed different bending behavior than that which was clamped because bending along the width direction is partially confined by the fixed edge. In addition to the effect of the clamp, the length-width ratio may also impact the degree of bending. In consideration of the length-width ratio, the bending deformation is more obvious along the long edge. Specifically, the radius of curvature ρ is the same if the swelling strain gradient is the same, and the bending angle $\beta = L/\rho$ is larger for long edge, where L is the length and β is the bend angle as shown in Fig. S10c. To more effectively highlight these two factors, in the manuscript, we have a larger length to width ratio for the sample clamped at one edge, so the bend along the width direction can be neglected. In the case of bilayer actuation, the bending direction was programmed during the actuator fabrication process. As mentioned in our manuscript and other reports of bilayer actuators, these actuators are comprised of two or more layers of materials in the cross-section direction which have different mechanical and/or chemical properties. Under the same stimulus, the two sides of the bilayer actuator have different swelling or contraction ratios which result in the actuation behavior. In the present case, the active layer, PFSA solution was sprayed onto the surface of a flexible inert polymer layer (polyethylene glycol terephthalate (PET)) which has poor moisture absorption performance.

As described in the manuscript, when humidity increased, swelling of the thinner PFSA layer resulted in spreading actuation behavior of bilayer film (Fig. S11d and Movie S3). Decreasing the humidity caused the bilayer film to recoil into its original state owing to the contraction of the PFSA layer.

The revision made to the Supplementary Information follows:

“Furthermore, we used both the experiments and simulations to interrogate the contribution of the effect of the clamped position and the structure of actuators (length-width ratio) to bending direction (Fig. S10).” (Page 10, Line 225~227)

Fig. S10 The experiments (a) and simulations (b) shown the contribution of the effect of the clamped position and the structure of actuators (length-width ratio) to bending direction. (c) The actuation curvatures versus length-width ratio of single-layer one edge fixed PFSA film.

“Here, for the single-layer PFSA actuator film, we used both experiments and simulations to interrogate the contribution of the effect of the clamped position and the structure of actuators (length-width ratio) to bending direction. As shown in Fig. S10a and b, the PFSA film unclamped actuator showed different bending behavior than that which was clamped because bending along the width direction is partially confined by the fixed edge. In addition to the effect of the clamp, the length-width ratio may also impact the degree of bending. In consideration of the length-width ratio, the bending deformation is more obvious along the long edge. Specifically, the radius of curvature ρ is the same if the swelling strain gradient is the same, and the bending angle $\beta = L/\rho$ is larger for long edge, where L is the length and β is the bend angle as shown in Fig. S10c. To clear highlight these two factors, in the manuscript, we have a larger length to width ratio for the sample clamped at one edge, so the bend along the width direction can be neglected.” (Page 13 of Supplementary Information, Line202~214)

Reviewer #2

In this manuscript, the authors demonstrated the vapor driven actuation of perfluorosulfonic acid ionomer (PFSA) and its utilization in three impressive applications: 2D/3D bilayer actuator, a wearable actuator for humidity/heat management, as well as interactive mechanochromic actuation with the integration of chromogenic photonic crystals. These applications are uniquely designed, well demonstrated and undoubtedly may enable several new applications. This manuscript should be considered for publication after the following modifications.

We are grateful to Reviewer 2 for recognizing the importance of this contribution to the scientific community and the insightful comments on our work.

1. Differentiation: PFSA film, such as DuPont NAFION (same chemical structure as presented in Figure 1A) is known for its nanochannel structure for ion transportation, and actuator applications, such as ionic polymer-metal composite (IPMC) has drawn many interests. Although the focus of IPMC is switching electrical signals to shape formation, the bending mechanism is similar and has been extensively studied. The authors should discuss those applications in the introduction session with a clear explanation of the novelty of the work presented in this manuscript.

Response: Great suggestions! The ionic polymer-metal composite (IPMC) and its applications in electrical signals and actuation have been discussed in the introduction section. In addition, the novelty/superiority of the current work in actuation has also been clarified in the Revised Manuscript. More details can be found in Page 4 in the Revised Manuscript.

The revision made in the revised manuscript:

“The use of PFSA, otherwise known as Nafion™ as an ion-exchange membrane in ionic polymer-metal composite (IPMC) artificial muscle has been reported^{28,29}. Actuation is induced by application of an electric field whereby water molecules within the membrane move between two metal electrodes. IPMC performance, however, is limited due to electrolysis of water and/or water evaporation from cracks in the electrodes. Long-term cycling also leads to adhesion failure at the sandwiched interfaces. Here, we take the advantage of the nanoscale molecular channels and vapor absorbing functional groups present in Nafion™ to fabricate a series of vapor driven actuators. The commercially available membrane material was studied as an intrinsically deformable and foundational material for a vapor-driven actuator. A series of vapor driven single-layer and bilayer actuators that exhibited a rapid response (up to 0.25 s for one curl) and high stability (>8,000 cycles without deterioration) were designed and fabricated by a simple and practical process while avoiding interface problems and long-range movement of molecules in the nano channel.” (Page 4, Line 90~101)

2. Monolayer PFSA actuation property experimental: For sensor/actuator application, reproducibility and selectivity are important figure of merits. I would recommend to present Fig. S5b, S6 in the main content, possibility in combination with Fig. 2.

In addition, from Fig. S5a, the PFSA membrane only experienced vapor exposure from one side. If this is true, the discussion of actuation performance simulation (supplement note 3, Fig. 3c, Fig. S7) is confusing as the vapor concentration on the opposite side of the film (c_L) is undefined and will continuously increase with time (i.e., not a steady state). The authors should elaborate more, possibility includes a diffusivity and time-dependent error function, in order to find better agreement between the simulation results (Fig. 3c, Fig. S7, up to 5% of swelling strain gradient) and the experimental results (Figure 2b and Fig. 3a as transient condition and Fig. S5 as pseudo-steady-state condition, up to 160 degree of bending angle which is much larger in magnitude compared to 5% of swelling strain gradient)

Response:

Thank you for the comments. According to the recommendation, we now present Fig. S5b, S6 in the main content (Fig. 2b insert and Fig. 3d). And for the simulation, we include a diffusivity and time-dependent error function to capture the diffusion behaviors from transient to steady-state, and discuss the pseudo-steady state condition as suggested.

The revision made in the revised manuscript and revised Supplementary Information follows:

“Consider the diffusion through the PFSA membrane with thickness L .

$$\frac{\partial c(x,t)}{\partial t} = D \frac{\partial^2 c(x,t)}{\partial x^2} \quad (1)$$

where c is the concentration function and D is the diffusion coefficient. Combined with the boundary conditions $c(x,t) = c_0$ at $x = 0$ and $c(x,t) = c_L$ at $x = L$, the transient solution of concentration profile for Eq. 1 is

$$c(x,t) = c_0 + \frac{c_L - c_0}{\operatorname{erf}\left(\frac{L}{\sqrt{4Dt}}\right)} \operatorname{erf}\left(\frac{x}{\sqrt{4Dt}}\right) \quad (2)$$

where $\operatorname{erf}(x)$ is the error function and is defined as $\operatorname{erf}(x) = \frac{2}{\sqrt{\pi}} \int_0^x e^{-x^2} dx$

The concentration profiles $c(x)$ within the membrane of thickness L are curved until steady state has been reached, as shown in Fig. S9a. The curvature takes time to dissipate, in accord with Fick's Second Law (Eq. 1), until the pseudo-steady state is reached at about $t_4 = L^2/2D$ in Fig. S9a, where the concentration profile $c(x)$ is approximated as $c_0 - (c_0 - c_L)x/L$. The swelling strain distribution $\varepsilon(x)$ along the thickness of the membrane is assumed to be proportional to the concentration profile $c(x)$ as,

$$\varepsilon(x) = \alpha c(x,t) \quad (3)$$

where α is the swelling coefficient. In the experiments, exposure of one side of a PFSA membrane (thickness $L = 75 \mu\text{m}$) to ethanol vapor led to bending of the film and the

maximum curvature was about 0.3 mm^{-1} (**Fig. S9b**), which can be seen in the pseudo-steady state, and the concentration profile $c(x)$ is approximated as $c^0 - (c^0 - c^L)x/L$. Thus the curvature κ in the pseudo-steady state can be expressed as,

$$\kappa = \alpha(c^0 - c^L)/L \quad (5)$$

To produce the bending actuation of the PFSA membrane with maximum curvature $\sim 0.3 \text{ mm}^{-1}$, the needed swelling strain gradient $\alpha(c_0 - c_L)$ is about 2.3%, validated from both FEA simulation and theoretical analysis. With these experimental parameters $\alpha(c_0 - c_L) = 2.3\%$, and **Eqs. 2-3**, a series of bending snapshots at $t = 0$ and t_1, t_2, t_3 , and t_4 were simulated by FEA, where the bending curvature at $t = t_4$ is about 0.3 mm^{-1} and agrees with the experimental results.” (Page 10 of Supplementary Information, Line146~173)

Fig. S9 (a) Transient concentration profile in diffusion across the membrane with thickness L . The pseudo-steady state is approached at t of about $L^2/2D$, diagrammed as t_4 , where the profile is an approximate straight line between $c(x,t) = c_0$ at $x = 0$ and $c(x,t) = c_L$ at $x = L$. (b) FEA simulation of a series of bending snapshots at $t = 0$ and t_1, t_2, t_3 and t_4 , where the bending curvature at $t = t_4$ is about 0.3 mm^{-1} . It should be noted that the simulation is one segment of a PFSA membrane with a length of 1.5 mm and thickness $75 \text{ }\mu\text{m}$. (in experiments the length was 15 mm)

Reviewers 3#

The authors present interesting results from research on vapor driven, polymeric thin film actuators. The novelty here is the demonstration that PFSA (Nafion) thin films, which intrinsically contain nanoscale molecular channels, can respond quickly to changes in the humidity or ethanol vapor concentration with large conformational changes. Nafion thin films have been used in many sensing and actuation applications, but typically are actuated by application of a potential across the film. What's novel and surprising here is the rather fast and repeatable actuation by gradients in RH or other polar (e.g., ethanol) vapors. To this reviewer at least, the demonstration of self-adaptive bilayer actuators and interactive mechanochromic actuation were scientifically more exciting. As such, the authors should consider to change the title of the paper to focus more on these applications (which are not targeted to human-environment interface applications), and as an additional feature, also discuss the "heat and moisture regulating" shirt. The paper is clearly written, and the data shown support the conclusions drawn. A few issues should be addressed however.

We are grateful to Reviewer 3 for recognizing the significance and novelty of our contribution to the scientific community. In addition, according to the great suggestions, the title of the paper has been changed to "Molecular-Channel Driven Actuation: Implications for Multi- Geometry/Color Switch", to focus more on the applications in self-adaptive bilayer actuators and interactive mechanochromic actuation. In addition, , added discussion about the "heat and moisture regulating" shirt has been provided in the Revised Manuscript and Supporting Information.

1. *As mentioned, consider shifting the focus of the paper toward the surprising conformational responses of PFSA due to changes in humidity and other polar vapors.*

Response: Thank you for the kind advice. Accordingly, the description and discussion about the surprising conformational responses of PFSA due to changes in humidity and other polar vapors have been augmented in the Revised Manuscript. In addition, more experiments and finite element analysis to further understand the actuation mechanism have also been incorporated.

2. *The authors may wish to call out the PFSA also by its commercial name Nafion.*

Response: Great suggestion! The commercial name Nafion™ has now been clearly presented in the revised manuscript.

3. *Page 3, Line 58: references should be added after each materials class.*

Response: Thank you! The references have been added after each materials class in Page 3, Line 58~61 in the Revised Manuscript.

4. *Page 6: The thickness of the films studied by GIWAXS should be given.*

Response: The thickness (232 ± 17 nm) of the films studied by GIWAXS has been provided in the caption of Figure 1 in the original manuscript. To better clarify this point to the readers, this information has been provided in the revised manuscript.

“To further investigate the morphology evolution upon exposure to vapor, *in situ* GIWAXS characterization was performed using ethanol as the polar medium (film thickness was controlled at ca. 232 ± 17 nm).” (Page 6, Line 128~130))

5. Page 8: Provide an explanation why methanol yielded the largest bending actuation. What do we know about the specific interactions of methanol with PFSA?

Response:

Thank you for this question. There are two factors that affect the bending actuation behavior for different kinds of vapors: (1) The polarity of the solvent molecules themselves (The amount of solvent absorbed, decreased with a decrease in solvent polarity); (2) The saturated vapor pressure (The saturation pressure impacts the maximum vapor content in the air). The polarity of solvents used in this work is: water > methanol > acetonitrile > ethanol > 1-propanol > 1-butanol > cyclohexane > petroleum ether (the detailed information regarding the polarity has been added as Table S1 in the Revised Supporting Information). Water and methanol should lead to a larger degree of swelling and bending actuation of the PFSA film. However, solvent vapor (rather than solvent liquid) was applied to drive the PFSA based actuator. Owing to the saturated vapor pressure of water being lower than other solvents (methanol, acetonitrile, ethanol e.g.), the bending angle of PFSA film driven by moisture is smaller than methanol. To demonstrate our explanation better, we also measured the bending actuation using solvent vapors with polarities that vary significantly. As shown in Fig. S6, it was found that the solvent vapor (for example methanol, ethanol, acetonitrile) with high polarity and high saturated vapor pressure elicited stronger actuation response. Solvent vapors (such as cyclohexane and petroleum ether) with low polarity and saturated vapor pressure exhibited lower actuation performance.

The revision made to the revised manuscript and revised Supplementary Information follows:

“Multi-responsive bending actuation with both moisture and alternative solvent vapors was demonstrated (Fig. 3d). We found that high polarity and high saturated vapor pressure solvent vapors (i.e., methanol, ethanol, acetonitrile) (Table. S2), that is, vapors that interact strongly with the nano channel surface, trigger the strongest actuation; solvent vapors (i.e., cyclohexane, petroleum ether) that have low polarity drive a much weaker bending response.” (Page 9, Line 209~213)

Table S2. Polarity and saturated vapor pressure of solvents.

Solvent	Polarity	Saturated vapor pressure (kPa)
---------	----------	--------------------------------

Methanol	6.6	16.8
Ethanol	4.3	8.5
1-Butanol	3.7	0.82
1-propanol	4	5.8
water	4.2	3.16
Acetonitrile	6.2	12.15
Cyclohexane	0.1	13.3
Petroleum ether	0.01	53.3

6. Page 12: *Was the standard terylene-based sports shirt also the same material used for the “actuated” sports shirt? The comparison between the shirts needs to be made more clear. For example, one could measure vapor diffusion (in absence of mechanical actuators), ideally it should be equal for both types of shirts to arrive at a reasonable comparison. This reviewer found this section the scientifically weakest, while the idea and implementation certainly are intriguing.*

Response: Thank you for these questions. As we have shown in Fig. 5f, both the actuator based sports shirt and standard teryleneTM-based sports shirt comprise two layers. The lining material (Textile 1) for the two kinds of shirts is the same (knitted porous polyester fabric as shown in Fig. 5d and f). The outside layer of the actuator based sports shirt and standard terylene-based sports shirt are semilunar patterned PFSA and polyester fabric, respectively. However, the actuator based sports shirt uses a completely different drying mechanism than the traditional polyester fabric based counterparts. The pre-designed semilunar pattern on the NafionTM film opens when the moisture level increases near perspiring skin, allowing better ventilation of moisture into the external environment. On the other hand, moisture is transported to the external surface of the traditional polyester fabric clothes by capillary effects and evaporates to the environment. The actuator based shirt has certain advantages by its biomimetic response that mimics sweat pores. Furthermore, in order to make the comparison between the shirts to be more clear, we performed water vapor transmission rate (WVTR) tests for the PFSA film (without any pattern) based shirt, PFSA film with semilunar patterns shirt and standard polyester fabric sports shirt at RH=90%. The data now presented in Figure 5d show that the water vapor transmission rate of the polyester fabric shirt is 145 g/m² per hour, a value that is higher than determined for the PFSA film (without any pattern) based shirt (97 g/m² per hour). Notably, the actuator based sports shirt demonstrated a substantially higher water vapor transmission rate of up to 237 g/m² per hour., Additional scientific-related discussions and experimental data about the “actuated” sports shirt have been added to the revised manuscript. Definitely, more experiments and further studies will be conducted in future work.

The revision made in the revised manuscript and revised Supplementary Information:

“Water vapor transmission rate (WVTR) tests were conducted on an unpatterned PFSA film based shirt, PFSA film shirt having semilunar patterns and a standard polyester fabric sports shirt at RH=90%. The data in Fig. 5d demonstrate that the standard polyester fabric shirt exhibited a higher WVTR than the unpatterned PFSA analog, 145 g/m² per hour vs. 97 g/m² per hour, respectively. The actuator based sports shirt exhibited a significantly higher WVTR, up to 237 g/m² per hour. Furthermore, a test that involved a man running 3 kilometers was also conducted. When wearing the PFSA-actuator-based shirt, the runner’s skin temperature was as much as 1.3 °C lower than that when wearing typical commercially available sportswear (Fig. 5e and f). Thus, the PFSA layer appears to enable venting between the skin and the external environment, thereby facilitating control of body moisture and skin temperature.” (Page 12, Line 295~304)

Figure

5 Personal humidity and heat management system. (a) Fabrication of the actuator array using a laser cutting and patterning system. (b) Schematic diagram of the homemade humidity chamber used for quantitative measurement of the humidity control ability of personal humidity and heat management system. The photo of the actuator array before and after self-adaptive actuation (Scale bar is 1 cm) (c) Plots of time dependent–humidity change within the chamber affixed with normal PET film and personal humidity and heat management system. (d) Water vapor transmission rate test of three type of PFSA films with semilunar patterns shirt, standard polyester fabric sports shirt, and PFSA film (without any pattern) based shirt. (e, f) A PFSA matrix with semilunar patterns able to curl in an outward direction was integrated into commercial sports shirt to develop clothing for personal humidity and heat management. Plots of the time dependent–skin temperature. (Scale bar is 1.5 cm)

REVIEWERS' COMMENTS:

Reviewer #1 (Remarks to the Author):

The authors have addressed most of comments. However, there are still some questions to be solved.

1.As a single-layer actuator, its direction of movement depends on which side of this film that was exposed to vapor. In this study, the direction of the film movement and vapor should be noted clearly in Figures and text. On the other hand, when vapor triggered on opposite side of this film, the direction of movement would be entirely different.

2.Whether the bending angle should decrease after being exposed to the vapor for a lone time without the special equipment in Fig S6 because of the homogeneous absorption of vapor.

Reviewer #2 (Remarks to the Author):

In the revised version of the manuscript, the authors carefully and thoroughly addressed the reviewers' comments, including extensive new experimental data and theoretical models to support their findings, along with additional references and elaborations to highlight the novelty of the presented work based on a rather conventional Nafion film. The manuscript is good for publication.

Reviewer #3 (Remarks to the Author):

The authors have provided satisfactory answers to this reviewer's questions. Furthermore, by responding to all issues raised also by other reviewers, the manuscript has now substantially improved. Publication can now be recommended.

Response to comments

Reviewer 1

The authors have addressed most of comments. However, there are still some questions to be solved.

- 1. As a single-layer actuator, its direction of movement depends on which side of this film that was exposed to vapor. In this study, the direction of the film movement and vapor should be noted clearly in Figures and text. On the other hand, when vapor triggered on opposite side of this film, the direction of movement would be entirely different.*

Response: We appreciate your valuable suggestions. According to the reviewer's suggestions, the direction of the film movement and vapor were noted in Figures and text. Based on the mechanism we explain in the manuscript, the bending directions of the actuators were attributed to the direction of the stimulus. As shown in Supplementary Figure 7c (i and ii), the single-layer PFSA actuator film always bent in the opposite direction to the stimulus owing to the asymmetric volume expansion of the two sides of the actuator film.

- 2. Whether the bending angle should decrease after being exposed to the vapor for a lone time without the special equipment in Fig S6 because of the homogeneous absorption of vapor.*

Response: Thank you for this question. The experimental apparatus shown in Supplementary Figure 6 is a sample set-up which make sure that there was a steady vapor concentration on one side of the PFSA actuator, and there was as little as possible vapor on the other side. Without the experimental apparatus shown in Supplementary Figure 6, to address the reviewer's comments, we used the experiments to interrogate the bending actuation behavior of the monolayer PFSA actuator under two situations in detail. The first one is the bending actuation of monolayer PFSA actuator when it was put in a homogeneous vapor atmosphere. As shown in Supplementary Figure 7c (iii), the monolayer PFSA actuator did not show obvious bending behavior because of the uniform vapor absorption on two sides of the PFSA film. On the other hand, the single-layer PFSA actuator exhibited obvious bending behavior when it was put in an ethanol vapor atmosphere which has a concentration gradient (Supplementary Figure 6c). As presented in Supplementary Figure 6d, the maximum bending angle is smaller compared to the bending actuation when the set-up in Supplementary Figure 6c was used. This was because the vapor concentration difference between two sides of single-layer PFSA actuator is small without the experimental apparatus (Supplementary Figure 6a).

The following revision was made to the manuscript and Supplementary Information:

“In addition, the bending actuation performance of single-layer PFSA film also can be affected by gravity and stimulus direction. Specifically as shown in Supplementary Figure 6d and 7c, the single-layer PFSA actuator film always bent in the opposite direction to the higher vapor concentration side owing to the asymmetric volume expansion of the two sides of the actuator film and the phenomenon is film thickness-dependent as shown in Fig. S8.” (Page 8, Line 179~182)

Supplementary Figure 6. Effect of ethanol vapor concentration on the curvature of monolayer PFSA membrane actuator. **a** Schematic illustration of the experimental measurement apparatus. **b** The scheme used for definition and calculation of curvature. (The three dashed circles (red, yellow and blue) represent the profile of the PFSA film. Here, “*r*” is the radius of every bent arch. The curvature was calculated by the equation below. $Curvature = 1/r$.) **c** Schematic illustration of the experimental measurement apparatus. **d** Single-layer PFSA actuator bending angle versus time when it was exposed in an ethanol vapor atmosphere which has concentration gradient.

Supplementary Figure 7. Bending curvature change of single-layer PFSA films with different thickness. **a** Plots showing time dependence of the bending curvature change of single-layer PFSA films with different thickness upon exposure to 18% ethanol vapor conditions and relaxation dynamics after removing the ethanol vapor exposure. **b** Comparison of bending actuation properties and the response time in our study and previous bending actuators.¹²⁻²³ **c** The bending actuation behavior of single-layer PFSA triggered in different directions (i) trigger from left side, (ii) trigger from right side, (iii) trigger from two sides of the actuator.

Reviewer 2

In the revised version of the manuscript, the authors carefully and thoroughly addressed the reviewers' comments, including extensive new experimental data and theoretical models to support their findings, along with additional references and elaborations to highlight the novelty of the presented work based on a rather conventional Nafion film. The manuscript is good for publication.

Response: We are grateful to Reviewer 2 for recognizing the importance of this contribution to the scientific community and support for the publication.

Reviewer 3

The authors have provided satisfactory answers to this reviewer's questions. Furthermore, by responding to all issues raised also by other reviewers, the manuscript has now substantially improved. Publication can now be recommended.

Response: We are grateful to Reviewer 3 for recognizing the publication of our manuscript.